# Genome-wide screen in human plasma identifies multifaceted complement evasion of *Pseudomonas aeruginosa*

**Manon Janet-Maitre[1], Stéphane Pont[1¤a], Frerich M. Masson[2], Serena Sleiman[1], Julian Trouillon[1¤b], Mylène Robert-Genthon[1], Benoît Gallet[3], Chantal Dumestre-Perard[3,4], Sylvie Elsen[1], Christine Moriscot[5], Bart W. Bardoel[2], Suzan H. M. Rooijakkers[2], François Cretin[1], Ina Attrée[1]***

**1** University Grenoble Alpes, Institute of Structural Biology, UMR5075, Team Bacterial Pathogenesis and Cellular Responses, Grenoble, France, **2** Medical Microbiology, University Medical Center Utrecht, Utrecht University, Utrecht, The Netherlands, **3** University Grenoble Alpes, CNRS, CEA, Institute of Structural Biology, Grenoble, France, **4** Laboratory of immunology, Grenoble Alpes University Hospital, Grenoble, France, **5** University Grenoble Alpes, CNRS, CEA, EMBL, ISBG, UMS3518, Grenoble, France

¤a Current address: Laboratory of Pathogen-Host Interactions (LPHI), Université Montpellier, CNRS, UMR5235, Montpellier, France
¤b Current address: Institute of Molecular Systems Biology, ETH Zurich, Zurich, Switzerland
* ina.attree@ibs.fr

## Abstract

*Pseudomonas aeruginosa*, an opportunistic Gram-negative pathogen, is a leading cause of bacteremia with a high mortality rate. We recently reported that *P. aeruginosa* forms a persister-like sub-population of evaders in human plasma. Here, using a gain-of-function transposon sequencing (Tn-seq) screen in plasma, we identified and validated previously unknown factors affecting bacterial persistence in plasma. Among them, we identified a small periplasmic protein, named SrgA, whose expression leads to up to a 100-fold increase in resistance to killing. Additionally, mutants in *pur* and *bio* genes displayed higher tolerance and persistence, respectively. Analysis of several steps of the complement cascade and exposure to an outer-membrane-impermeable drug, nisin, suggested that the mutants impede membrane attack complex (MAC) activity *per se*. Electron microscopy combined with energy-dispersive X-ray spectroscopy (EDX) revealed the formation of polyphosphate (polyP) granules upon incubation in plasma of different size in *purD* and wild-type strains, implying the bacterial response to a stress signal. Indeed, inactivation of *ppk* genes encoding polyP-generating enzymes lead to significant elimination of persisting bacteria from plasma. Through this study, we shed light on a complex *P. aeruginosa* response to the plasma conditions and discovered the multifactorial origin of bacterial resilience to MAC-induced killing.

## Author summary

Persistence of bacterial pathogens is a main cause of treatment failure and establishment of chronic bacterial infection. Despite innate immune responses, some bacteria may

**Data Availability Statement:** The authors confirm that all data underlying the findings are fully available without restriction. All relevant data are either within the paper and its Supporting Information files or available through the NCBI Gene Expression Omnibus (GEO) under super-series accession number GSE192831, or by using accession numbers GSE192769 and GSE192761 for the Tn-seq and RNA-seq complete data sets, respectively.

**Funding:** The work described in this paper was supported by grants from the French national agency for research (Agence Nationale de la Recherche; ANR-15-CE11-0018-01), the Laboratory of Excellence GRAL, funded through the University Grenoble Alpes graduate school (Écoles Universitaires de Recherche) CBH-EUR-GS (ANR-17-EURE-0003), the Fondation pour la Recherche Médicale (Team FRM 2017, DEQ20170336705) to I.A., and the European Union's Horizon 2020 research programs H2020-EU-ITN-EJD (CORVOS No #860044 to F.M. and S. H.M.R.). This work availed of the platforms at the Grenoble Instruct-ERIC center (ISBG; UAR 3518 CNRS-CEA-UGA-EMBL) within the Grenoble Partnership for Structural Biology (PSB), supported by FRISBI (ANR-10-INBS-0005- 02) and GRAL, funded through the University Grenoble Alpes graduate school (Ecoles Universitaires de Recherche) CBH-EUR-GS (ANR-17-EURE-0003). S.P, J.T and M.JM were recipients of Ph.D. fellowships from the French Ministry of Education and Research. S.S received the Master 2 GRAL fellowship. The funders had no role in study design, data collection and analysis, decision to publish or preparation of the manuscript.

**Competing interests:** The authors have declared that no competing interests exist.

persist in human blood and plasma. Here we used a genome-wide screen to investigate the molecular determinants influencing *Pseudomonas aeruginosa* survival in human plasma facing the complement system. Alongside a multifactorial strategy that include surface-attached molecules and bacterial adaptation to stress, we found that intracellular polyphosphates and biotin significantly influence bacterial capacity to deal with membrane attack complex (MAC)-dependent killing. These results underline the need to understand the complex interplay between bacterial pathogens and the human immune system when seeking to develop efficient antibacterial strategies.

## Introduction

Human bacterial pathogens employ sophisticated strategies to escape control by the immune system and to resist or tolerate antibiotic treatments. Bacterial persistence toward antibiotics is a major obstacle in the treatment of life-threatening infections. It is frequently associated with the establishment of chronic infections, as it allows the survival of a subpopulation of bacteria that are tolerant to otherwise lethal antibiotic stress [1,2]. Unlike antibiotic resistance, persistence is a non-heritable, fully reversible trait characterized by a biphasic killing curve–with rapid killing of the bulk population and survival of a subpopulation over a long period of time. Among the factors hypothesized to lead to antibiotic persistence [3], the most cited are low intracellular ATP levels [4,5] and production of the "alarmone" signaling molecule guanosine (penta) tetraphosphate, (p)ppGpp [6,7]. The host immune system also elicits persistence in a number of bacterial species. For example, acidification of macrophage vacuoles after *Salmonella* internalization induces persister cell formation and could contribute to establishing a bacterial reservoir for infection relapse [8]. In a similar manner, exposure to human serum triggers the formation of antibiotic persisters as well as so-called "viable but non-culturable" forms of *Vibrio vulnificus* [9]. Putrinš *et al.* [10] reported that phenotypic heterogeneity in *Escherichia coli* could serve as a mean to evade serum-mediated and antibiotic-induced killing. In a previous study [11], we demonstrated that a persister-like sub-population of *Pseudomonas aeruginosa* evaders forms when generally sensitive bacteria are incubated in human blood or human plasma. In this article, we refer to resistance, tolerance, and persistence as defined in a context of exposure to antibiotics [3]. Briefly, resistance is due to a heritable resistance factor allowing survival in the face of higher stress conditions. In contrast, tolerance is a phenomenon that allows survival of the population despite an otherwise lethal stress for a longer period of time, but without deploying any resistance mechanism. The third term, persistence, is the capacity of a tolerant sub-population to survive whereas the rest of the population is eliminated. Finally, we used the term resilience to describe several of these phenomena.

*P. aeruginosa* is a leading nosocomial pathogen that is extremely difficult to eradicate due to its high adaptability to a range of environments and its intrinsic and acquired antibiotic resistance [12]. Bloodstream infections (BSI) due to *P. aeruginosa* strains hold the record for highest mortality rate, at about 40% [13,14]. The complement system (CS) is a complex multiprotein cascade and a major innate immune component in human blood responsible for Gram negative bacterial killing. The complement cascade can be activated through three pathways (classical, lectin and alternative) which all converge to the formation of the C3 convertase. C3 proteolysis by the C3 convertase releases C3b, which opsonizes the bacterial surface, in turn forming the C5 convertases. C5b and C6 interaction initiate the binding of C7 and C8 at the membrane [15]. The final step is the polymerization of multiple copies of C9 to form a C5b-9 pore in the bacterial outer membrane [16]—the membrane attack complex (MAC)—which leads to inner membrane damage and bacterial lysis [17].

The CS, which is the main immune component of blood responsible for bacterial elimination, is generally efficient against *P. aeruginosa* sensitive strains. However, a tiny fraction of the initial bacterial population can persist despite prolonged incubation in blood or plasma [11]. The kinetics of *P. aeruginosa* killing by human plasma displays a classical persister-like biphasic profile [3], with rapid elimination of up to 99.9% of the population, reaching a plateau at an average of 0.1% survival after 2 h. In our previous study, we found that, out of 11 BSI isolates tested, 3 were plasma-resistant, 3 displayed a tolerant phenotype, and 5 were able to form evaders. The formation of evaders may represent a frequent strategy used by *P. aeruginosa* to escape MAC-mediated bactericidal activity in plasma. The plasma evaders were distinct from antibiotic persisters since a stationary phase population did not give rise to a higher population of evaders, as reported for antibiotic persisters [18].

For this study, we employed a genome-wide approach to obtain insights into *P. aeruginosa* factors impacting levels of persistence in human plasma. A screen based on transposon-insertion sequencing (Tn-seq) was applied to a recent clinical isolate that was sensitive to plasma but produced between 0.01% and 0.1% of evaders after 6 h in plasma. We hypothesized that transposon mutants for which an increased proportion of evaders was detected or mutants with overall increased plasma tolerance/resistance could reveal novel determinants of bacterial resilience to plasma and pathogen immune escape. In addition to *P. aeruginosa* factors known to be involved in resistance to CS, such as O-specific antigen (OSA) length and exopolysaccharides, we identified an uncharacterized small periplasmic protein that increases complement tolerance up to 100-fold. Our screen and the associated phenotypic characterization of mutants also revealed that biotin, ATP, and polyphosphate (polyP) levels modulate pathogen sensitivity to MAC-induced lysis.

## Results

### Gain-of-function screen in plasma

In our previous study, we found that the *P. aeruginosa* isolate IHMA879472/AZPAE15042 (IHMA87) persisted in human blood and plasma. Although the overall population was mostly sensitive to complement killing, approximately 0.1% of evaders survived prolonged incubation in plasma [11].

To search for mutants generating higher numbers of evaders using a genome-wide approach, we constructed a transposon library of about 300,000 mutants in IHMA87 using the Himar-1 transposon. The transposon library was challenged in native human plasma (output) or heat-inactivated plasma (HIP, input)—devoid of complement activity—for three hours before harvesting (Fig 1A). Bacterial survival was expressed as colony forming unit (CFU) counts for the whole library before and after the challenge. This analysis revealed that overall survival was increased by about 2-log in the mutant library compared to the parental strain. Thus, the screen selected for mutants with increased survival. Resistant, hyper-evader, and tolerant phenotypes could be differentiated based on killing kinetics of isolated mutants during longer incubation in plasma.

Tn-seq data was analyzed both for insertions in coding regions and in intergenic regions, as the transposon included an outward *tac* promoter, which could modulate the expression of neighboring genes. In both analyses, transposon insertions strongly increasing bacterial survival were associated with three main pathways: biosynthesis of purine and biotin; and biosynthesis, production, and regulation of the alginates (Fig 1B and Table 1, full results in S1 Table). In addition, several novel determinants worth exploring were identified (Table 1).

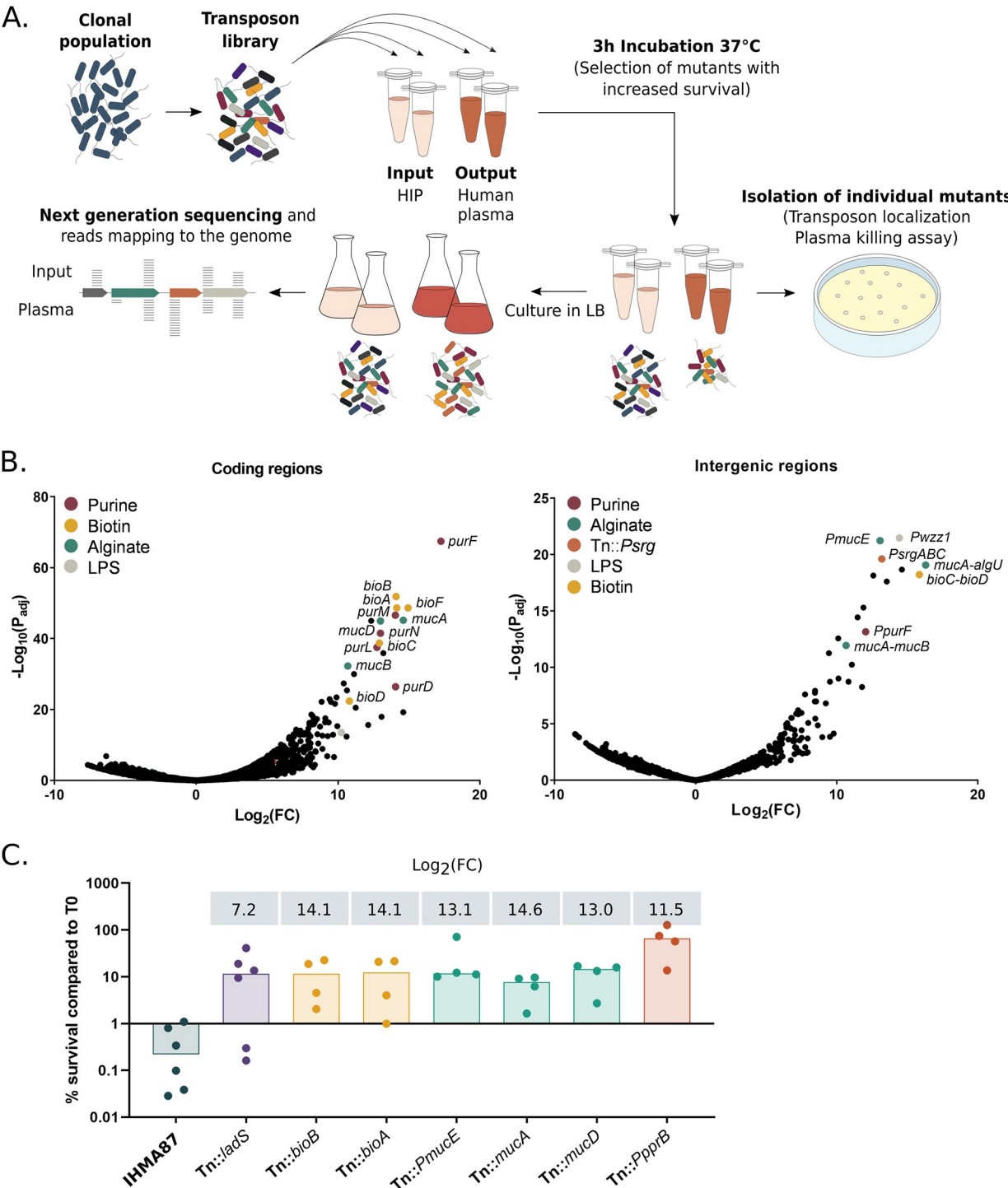

**Fig 1. Gain-of-function Tn-seq reveals common and novel pathways contributing to *P. aeruginosa* survival in plasma. A.** Schematic representation of the screening method used for this study. A transposon-insertion mutant library was generated in the plasma-sensitive clinical isolate *P. aeruginosa* IHMA87 and exposed either to human plasma or to HIP (input) for 3 h. Aliquots were plated on LB and individual transposon mutants were stored for further analysis. **B.** Bioinformatics analyses presented as Volcano plots. Insertions in coding regions (left) and intergenic regions (right) were analyzed separately. Significant hits in genes involved in the same pathway are shown in the same color. **C.** Survival of isolated transposon mutants following incubation for 3 h in plasma. Survival rates were calculated based on CFU measurements. The dataset was log-transformed, and rates for all mutants were significantly different from rates for the wild-type strain (*p*-value < 0.005). Log₂(Fold-Change (FC)) obtained from the Tn-seq analysis for each gene or intergenic region is indicated above the histogram.

**Table 1. Mutants enriched in plasma compared to HIP carry insertions in three main pathways.**

| IHMA87 ID [*] | PAO1 ID [**] | gene/prediction | Log$_2$(FC) [#] | P$_{adj}$ [##] |
|---|---|---|---|---|
| | | Purine biosynthetic pathway [$] | | |
| IHMA87_01837 | PA3108 | *purF* | 17.26 | 3.64E-68 |
| IHMA87_05343 | PA4855 | *purD* | 14.06 | 3.85E-27 |
| IHMA87_04291 | PA0945 | *purM* | 14.04 | 2.51E-47 |
| IHMA87_04292 | PA0944 | *purN* | 12.99 | 3.14E-42 |
| IHMA87_01191 | PA3763 | *purL* | 12.75 | 3.02E-38 |
| IHMA87_04214 | PA1013 | *purC* | 7.67 | 1.23E-5 |
| IHMA87_02487 | PA2629 | *purB* | 5.35 | 7.23E-4 |
| | | Biotin biosynthetic pathway | | |
| IHMA87_00477 | PA0501 | *bioF* | 14.94 | 2.29E-49 |
| IHMA87_00400 | PA0420 | *bioA* | 14.15 | 2.29E-49 |
| IHMA87_00476 | PA0500 | *bioB* | 14.10 | 1.39E-52 |
| IHMA87_00479 | PA0503 | *bioC* | 12.92 | 1.90E-39 |
| IHMA87_00480 | PA0504 | *bioD* | 10.80 | 4.23E-23 |
| | | Alginate biosynthetic/regulation genes | | |
| IHMA87_04493 | PA0763 | *mucA* | 14.60 | 6.76E-46 |
| IHMA87_04494 | PA0762 | *algU* | 11.25 | 3.01E-21 |
| IHMA87_00895 | PA4033 | P$_{mucE}$ | 13.07 | 5.88 E-22 |
| IHMA87_04490 | PA0766 | *mucD* | 13.00 | 1.10E-45 |
| IHMA87_04492 | PA0764 | *mucB* | 10.69 | 5.56E-33 |
| IHMA87_05847 | PA5322 | *algC* | 2.66 | 4.76 E-2 |
| | | Other top hits | | |
| IHMA87_02894 | NA | -/TraX family protein | 14.60 | 6.18E-20 |
| IHMA87_02666 | PA2484 | -/transcriptional regulator | 13.20 | 1.33E-36 |
| IHMA87_01574_IG | PA3368_IG | P$_{srg}$ | 13.19 | 2.50E-20 |
| IHMA87_01786_IG | PA3161_IG | P$_{wzz1}$ | 12.59 | 7.33E-19 |
| IHMA87_06317 | NA | - | 12.34 | 1.02E-45 |
| IHMA87_02554 | PA2573 | -/chemotaxis transducer | 11.13 | 1.02E-30 |
| IHMA87_06033 | PA5490 | *cc4* | 10.63 | 4.10E-26 |
| IHMA87_04274 | PA0960 | *slyX* | 10.60 | 4.67E-13 |
| IHMA87_04133 | PA1078 | *flgC* | 10.40 | 4.59E-28 |
| IHMA87_04779 | PA4457 | *kdsD*/arabinose-5-phosphate isomerase | 10.24 | 3.15E-14 |
| IHMA87_00319 | PA0341 | *lgt* | 9.95 | 5.21E-16 |
| IHMA87_00896 | PA4032 | -/probable response regulator | 9.89 | 3.67E-24 |
| IHMA87_03447 | PA1758 | *pabB* | 9.79 | 2.55E-22 |
| IHMA87_05840 | PA5315 | *rpmG* | 9.71 | 2.60E-07 |
| IHMA87_03022 | PA2122 | -/lipid biosynthetic process | 9.65 | 1.27E-13 |
| IHMA87_01755 | NA | - | 9.51 | 7.37E-23 |
| IHMA87_04617 | PA4315 | *mvaT* | 9.41 | 1.21E-23 |
| IHMA87_02603 | NA | - | 9.34 | 4.46E-17 |
| IHMA87_02570 | NA | - | 9.28 | 3.29E-13 |
| IHMA87_05344 | PA4856 | *retS* | 9.11 | 2.94E-18 |

[*] IHMA87 gene identity (ID) (NCBI accession numbers CP041354 and CP04135, for the chromosome and plasmid sequences respectively [41]) are indicated with PAO1 equivalents [**] where applicable. IG is added when insertions are in the intergenic region between two genes and the intergenic region is attributed to the upstream gene, independently of the gene's orientation.

[#] For each gene, the Log$_2$(FC) in plasma compared to HIP (input) condition is indicated together with the corresponding adjusted *p*-value (P$_{adj}$ [##]), obtained by the DESeq2 program.

[$] The genes are sorted according to their predicted pathway taken from the KEGG database and the *Pseudomonas* genome database [42,43].

## Analysis of Tn mutants isolated during the screen

In parallel to Tn-seq, ten transposon mutants were isolated during the screen, targeting seven different genes, and their survival was assayed in plasma killing assays. Survival rates for these mutants were between 10- and 200-fold higher than that of the parental strain (Fig 1C), thus validating the screen. Of ten mutants, three mutants displayed hyper-mucoid phenotypes–reflecting alginate overproduction–associated with insertions within *mucD* and in the intergenic region upstream *mucA*. *P. aeruginosa* can produce three main exopolysaccharides; alginates, Psl and Pel [19,20]. Expression of the alginate biosynthetic genes is regulated by the extracytoplasmic function (ECF) sigma factor AlgU. In planktonic growth, AlgU is retained at the bacterial inner membrane by its anti-sigma factor MucA, repressing alginate production [21]. MucD acts upstream of MucA and indirectly acts as a negative regulator of the release of AlgU in response to the presence of misfolded outer membrane proteins [22]. We also isolated a mutant in which the transposon was inserted into the intergenic region upstream of *mucE*, encoding a positive regulator of alginate production [23,24]. The orientation of the transposon and its mucoid phenotype suggested an overexpression of *mucE*. As previously reported, alginate overproduction may interfere with the CS by decreasing C3b-dependent opsonization and thus preventing CS activation upon alginate acetylation [25,26].

One of the mutants with increased survival carried a transposon insertion in *ladS*. LadS is a part of a signaling pathway RetS/LadS/Gac/Rsm regulating Psl exopolysaccharides production. This complex regulatory system which manages *P. aeruginosa* phenotypic switch from planktonic to biofilm lifestyle is composed of a central two-component regulatory system GacA/GacS modulated by two hybrid sensor kinases, LadS, a positive regulator of the pathway, and RetS, which represses the system. When activated, the GacA/GacS two-component regulatory system represses the Psl production [27–30]. In accordance with our data, LadS-regulated exopolysaccharide Psl were reported to improve bacterial survival of a mucoid strain in serum [31]. However, the role of Psl in complement resistance might be strain-dependent, as assays using the reference strain PAO1 initially showed that Psl reduces bacterial opsonization without affecting C9 insertion or bacterial survival in serum [32].

Survival in plasma was also increased by insertion of the transposon in the promoter of *pprB* (survival ~50%, Fig 1C). PprB is a regulator acting on type IVb pili/Tad, BapA adhesin and CupE fimbriae expression [33,34], suggesting that type IVb pili or CupE fimbriae could be also involved in *P. aeruginosa* plasma resilience. As shown in *Vibrio cholerae* and *Neisseria gonorrhoeae*, type IV pili contribute to serum resistance by recruitment of the host negative complement regulator C4BP [35,36].

Although none of the ten randomly picked colonies had a transposon insertion in LPS-related genes, the screen did confirm the importance of LPS and more specifically OSA biosynthesis in the bacteria's interaction with plasma with enrichment of mutants with Tn insertion upstream of *wzz*1 ($P_{wzz1}$, $Log_2(FC)$ of 12.59, Table 1). Wzz1 is one of the OSA chain length regulators [37–40]. Finally, in four isolated mutants with increased survival, the transposon was inserted into *bioA* and *bioB*, encoding enzymes involved in biotin biosynthesis (~12% survival each, Fig 1B). To our knowledge, these genes have not previously been associated with plasma or complement resistance.

## Expression of the three-gene operon *srgABC* improves survival in plasma

We focused first on a mutant carrying the insertion in an intergenic region between a three-gene operon *IHMA87_01573-IHMA87_01571* of unknown function, (*PA3369-PA3371* in PAO1, renamed *srgABC*) and *panM* (*IHMA87_01574/PA3368*), coding for a probable acetyltransferase (Fig 2A). This mutant was significantly enriched in Tn-seq datasets ($Log_2(FC)$ of 13.2; Fig 2B and Table 1). The mutant isolated during the screen displayed a tolerant

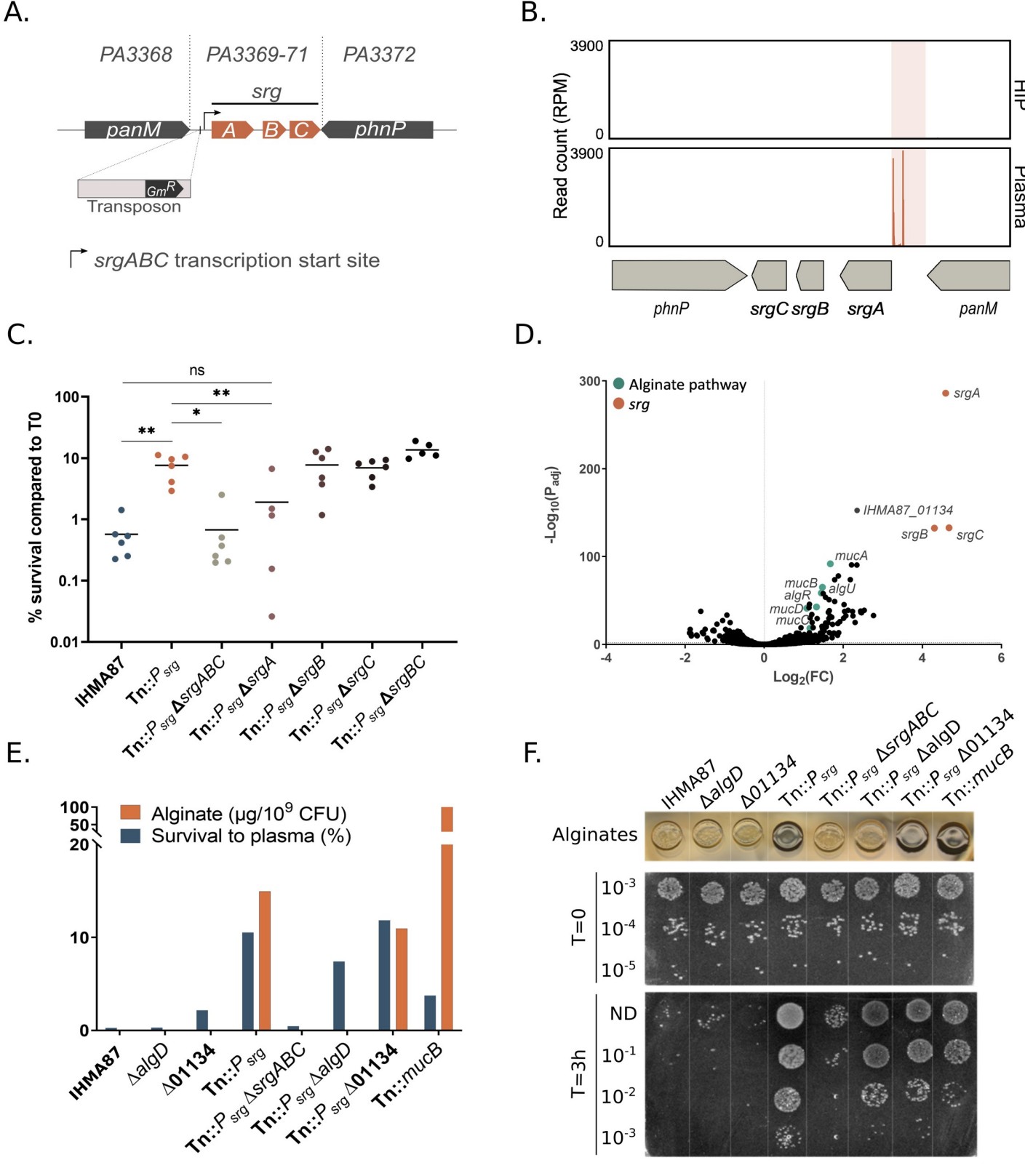

**Fig 2. Overexpression of *srgA* increases survival 100-fold. A.** Schematic representation of the predicted *srgABC* operon and position of the transposon in the isolated Tn::P*srg* mutant. Corresponding genes from PAO1 are indicated. Note that the transposon is inserted 13 bp upstream of the predicted transcriptional start site. **B.** Zoom in on the Tn-seq profiles in the region surrounding the *srg* operon, showing the number of normalized reads in input (HIP) and output (plasma). **C.** Role of individual *srg* genes in the plasma resilience phenotype of the Tn::P*srg* strain. Deletion of *srgA* restores sensitivity to plasma. **D.** Differential gene expression between Tn::P*srg* and the parental IHMA87 strain represented in a volcano plot showing overexpression of *srgABC* and alginate-related genes. RNA was extracted from cultures grown on LB, and whole transcriptomes were determined by an RNA-seq pipeline **E-F.** Survival in plasma and alginate production for *P. aeruginosa* strains. Alginate synthesis was visualized based on colony morphology and quantified by carbazole assay, as described in Materials and Methods. Alginate overproduction appears as a darker bacterial spot (**F**). The data shown are from one representative experiment, performed in biological triplicates.

phenotype over a 6-h period (S1 Fig). In addition, its survival was increased up to 100-fold compared to the parental strain after 3 h (Fig 2C). In preliminary experiments, upon exposure to plasma, survival of strains bearing deletions in *panM* or in *srgABC* was identical to survival of the wild-type strain. Therefore, to get an idea of the overall transposon-induced changes in the mutant, we determined its transcriptomic profile and compared it to the parental strain in LB. RNA-seq data (S2 Table) revealed overexpression (about 20-fold compared to the parental strain) of *srgABC* (for serum resistance genes). Based on transposon position upstream of *srgA*, the mutant will hereafter be referred to as Tn::P*srg*. Other genes were overexpressed in Tn::P*srg* compared to the wild-type strain (S2 Table), including *algU*–encoding the ECF sigma factor AlgU—and five other alginate genes (*algR* and *mucABCD* operon, Fig 2D).

To establish the relationship between the *srg* operon, alginate production and bacterial resilience to complement, we inactivated *algD*, encoding GDP-mannose 6-dehydrogenase, a checkpoint in alginate biosynthesis [44–46] both in the parental strain and in Tn::P*srg*, and investigated relevant phenotypes. In the assay, we included the alginate-overproducing mutant Tn::*mucB* isolated during the screen. We also designed a deletion mutant in the gene *IHMA87_01134* (*PA3819*), an uncharacterized gene that was overexpressed (Log$_2$(FC) = 2.35, P$_{adj}$ = 2x10$^{-153}$) in the Tn::P*srg* strain. For each strain, we established the survival rate by CFU counting, the quantities of alginates using carbazole assay and assessed the colony morphology on plates (Fig 2E and 2F). Tn::P*srg* displayed 10% survival in human plasma (0.01% for the parental strain), along with an increased alginate production. Although alginate overproduction can lead to higher resistance to human plasma, as seen for the mutant Tn::*mucB*, the alginates in Tn::P*srg* did not account significantly for its resistance in plasma. Finally, the deletion of IHMA87_01134 had no impact neither on alginate production nor on plasma resistance.

The *srg* operon encodes three small putative proteins: SrgA (9.9 kDa), SrgB (5.3 kDa) and SrgC (6.5 kDa). Structural predictions indicate SrgA to be a periplasmic protein with a signal peptide cleavage site at position Gly27. Both SrgB and SrgC are predicted to be membrane proteins with one and two transmembrane alpha-helices, respectively. All three *srg* gene products are highly conserved (98–100% amino acid sequence identity) over the 232 complete *P. aeruginosa* genomes available in the *Pseudomonas* genome database [43]. The individual contributions of the *srg* genes to *P. aeruginosa* tolerance to plasma were evaluated by testing single, double and triple deletion mutants directly in the Tn::P*srg* background strain. Plasma killing assays performed with these deletion mutants showed that SrgA was required and sufficient for increased survival in plasma (Fig 2C).

Overall, these results demonstrate that a *srg* overexpressing mutant produces more alginates and has a drastically increased plasma resistance in an alginate-independent manner and that SrgA alone is sufficient to modulate the plasma resistance.

## Purine and biotin pathway deficiencies lead to increased persistence

We then focused on mutants for which the transposon was present in genes and intergenic regions of biotin and purine biosynthesis. Mutants with insertions in the *bioBFHCD* operon and in the *bioA* gene–covering all steps of biotin biosynthesis–were significantly overrepresented following plasma challenge (Table 1 and Fig 3A, top). This result

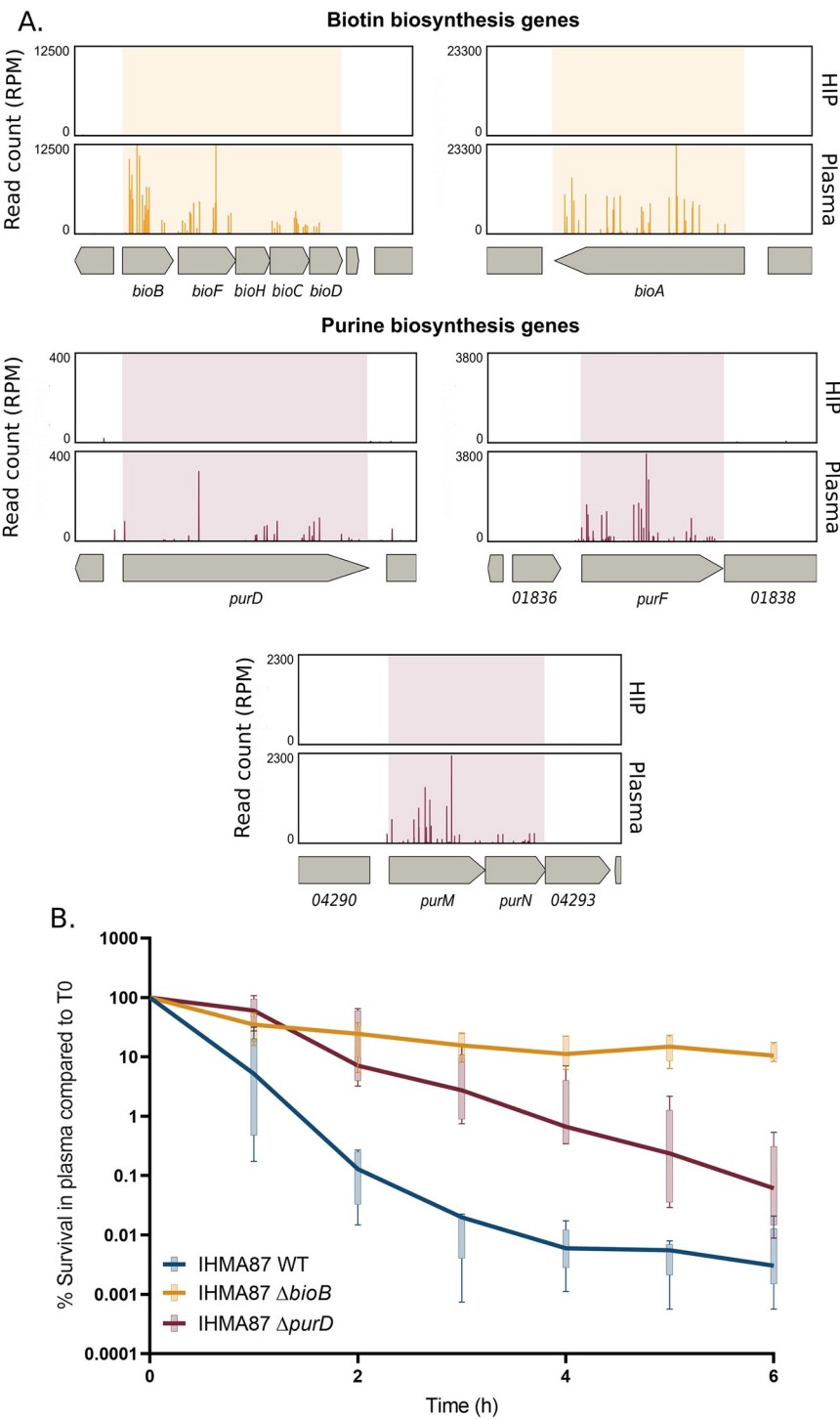

**Fig 3. Inactivation of biotin and purine biosynthetic pathways increases survival rates of IHMA87 in plasma. A.**
Zoom in on Tn-seq profiles of *bio* and *pur* genes and operons, showing normalized numbers of reads in input (HIP)
and output (plasma) samples. **B.** IHMA87 wild-type strain, Δ*bioB*, and Δ*purD* mutants' survival kinetics in plasma
over 6 h incubation, as measured by CFU counting (n = 5). Note the biphasic killing curves for the parental strain and
Δ*bioB* mutant, indicating increased persistence of Δ*bioB*. The Δ*purD* mutant displayed increased tolerance compared
to the parental strain.

suggests that lack of biotin promotes survival in plasma. This finding was unexpected as the biotin pathway has been shown to be essential for survival in human serum for several pathogens, including *Klebsiella pneumoniae* and *Mycobacterium tuberculosis* [47–49]. To confirm this result, in addition to the transposon mutants isolated from the screen (Fig 1C; Tn::*bioA* and Tn::*bioB*), we engineered a mutant carrying a deletion of *bioB* (Δ*bioB*), coding for the biotin synthase and examined its behavior in plasma over a 6-h challenge (Fig 3B). The Δ*bioB* mutant survived significantly better than the wild-type strain in plasma (over 10% *vs.* about 0.01% survival, respectively). Moreover, in plasma, IHMA87Δ-*bioB* displayed a biphasic killing curve, implying that the levels of biotin determine the persistence levels.

Like biotin synthesis, survival in plasma was clearly enhanced for bacteria bearing mutations in the purine biosynthetic pathway (Fig 3A, bottom). *De novo* purine biosynthesis, together with pyrimidine production, are essential for bacterial nucleotide metabolism and important for bacterial growth and survival in plasma and serum [50–53]. Transposon-insertion mutants in three main operons encoding enzymes involved in purine biosynthesis were overrepresented in the screen (Table 1). The two first hits in the pathway *purF*, encoding an amido-phospho-ribosyltransferase, and *purD*, encoding phosphoribosylamine-glycine ligase, were significantly enriched in plasma with a $Log_2$(FC) of 17.26 and 14.06, respectively (Table 1). To further study the role of purine biosynthesis in plasma resistance, we designed a deletion mutant of *purD* (Δ*purD*), given its non-operonic structure. Killing kinetics of Δ*purD* was studied over 6 h in plasma. In accordance with Hill *et al.* in an antibiotic context [54], the Δ*purD* mutant displayed a tolerant phenotype in plasma (Fig 3B). Interestingly, no similar level of gene enrichment was found for the pyrimidine pathway suggesting that the specific product of the purine pathway, rather than the nucleotide metabolism as a whole, contributed to increased survival in human plasma.

## ATP levels influence *P. aeruginosa* tolerance to plasma

Recent studies showed that low intracellular ATP concentration resulted in increased antibiotic persistence [4,5]. Along the same line, bacteria harvested at the stationary growth phase have lower levels of intracellular ATP and form a higher proportion of persisters when treated with antibiotics [55]. As the ATP molecule is a final product of the purine pathway (Fig 4A), we hypothesized that it could also play a role in the emergence of evaders. To investigate this hypothesis, we first measured intracellular ATP levels in Δ*purD* both in rich medium and after incubation in HIP. As expected, in both media, the ATP concentration measured for Δ*purD* was 5-fold lower than that measured for the parental strain (Fig 4B). To confirm that the low ATP concentration was indeed responsible for higher resilience to human plasma, we modulated ATP levels by supplementing the plasma with exogenous ATP during the plasma killing assay (Fig 4C). The addition of ATP to a concentration—in the range of that in healthy individual blood (Human Metabolome database [56])—when bacteria were exposed to plasma restored a wild-type-like sensitivity, further demonstrating the ATP-dependent phenotype of *pur* mutants.

No apparent links between the biotin biosynthesis pathway (Fig 4D) and ATP synthesis exist. Nevertheless, as biotin is involved in many biological processes, we tested whether the Δ*bioB* mutant could be rescued by ATP. As shown in Fig 4C, biotin but not ATP supplementation restored a wild-type-like sensitivity to Δ*bioB*, suggesting a distinct molecular mechanism for evader formation in this mutant. A double mutant *purD-bioB* exhibited an even higher survival rate in plasma than both simple mutants, further showing that the two pathways are independent and important in bacterial strategy to evade the CS (S2 Fig).

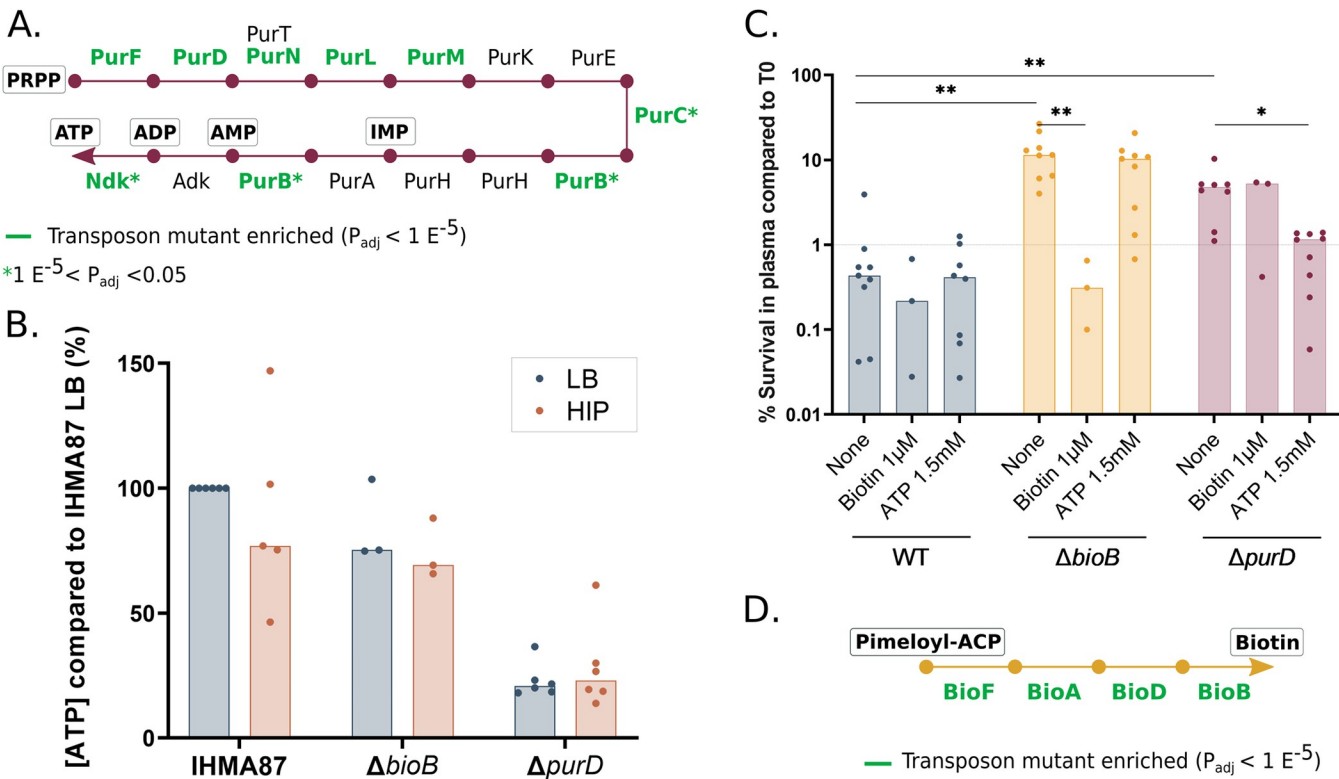

**Fig 4. ATP and biotin influence bacterial sensitivity to plasma. A.** Schematic view of the purine pathway (adapted from KEGG database [42]). Significantly enriched insertions in corresponding genes of the Tn-seq screen are indicated. **B.** Measurement of intra-bacterial ATP levels in LB and after 2h-incubation in HIP normalized to the CFU counts. **C.** Trans-complementation of Δ*bioB* and Δ*purD* phenotype by exogenous biotin or ATP, respectively. Biotin (1 μM) and ATP (1.5 mM) were added at the beginning of the incubation of bacteria in plasma. The survival was estimated by CFU counting and the median of all independent experiments is represented by the histogram. Statistical analysis was performed and p-value <0.05 or 0.01 are indicated with '*' and '**', respectively. **D.** The biotin biosynthetic pathway. Significantly enriched hits in Tn-seq are highlighted in green.

### Formation of energy-storage polyP granules in response to plasma

To further investigate the relationship between ATP and plasma resilience, we measured the ATP concentration in plasma. Compared to LB where we could not detect any ATP, different plasma pools contained between 15 and 45 nM ATP (S3 Fig), suggesting that bacteria do not encounter an ATP-depleted environment. As in addition to ATP depletion, the purine starvation engenders a stress response to bacteria [57], we investigated potential physiological and/or morphological changes that could explain the increased survival of the Δ*purD* mutant. Transmission electron microscopy (TEM) images were acquired for the different mutants either grown in LB medium or after plasma challenge (Fig 5A). In LB, the parental strain and the mutants were of similar size, shape and had similar apparent envelope thickness. Wild-type and Δ*bioB* bacteria incubated in human plasma systematically presented electron-dense granules of variable sizes (Figs 5A and S4). The Δ*purD* mutant cells occasionally harbored small electron-dense granules even in LB (Fig 5A). The size and number of the electron dense granules increased upon incubation of Δ*purD* in human plasma (Fig 5A).

Literature mining indicated that those electron-dense structures could be polyP granules. To cope with an array of different stresses, including nutritional and oxidative stress, *P. aeruginosa* produces long polyP chains from both GTP and ATP [58,59]. PolyP are then packed into granules to serve as energy and phosphate storage [58,60]. The granules detected here were confirmed to be phosphorus- and oxygen-rich structures by energy-dispersive X-ray

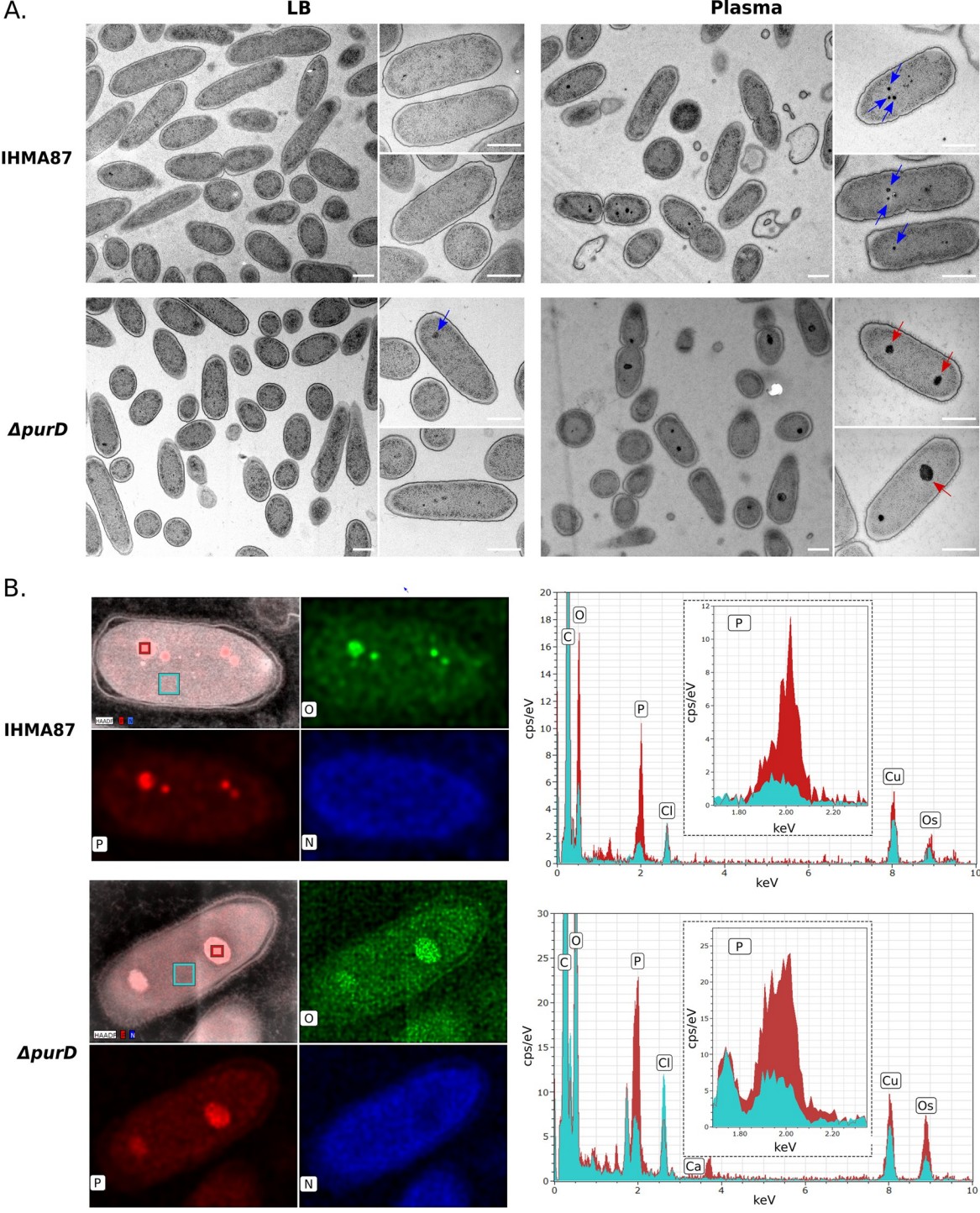

**Fig 5. Polyphosphate granules are formed upon *P. aeruginosa* incubation in plasma. A.** Transmission electron microscopy images of indicated strains after growth in LB (left) or 1h-incubation in human plasma (right). Red and blue arrows show big- and small-sized granules respectively. Scale bar = 500 nm. **B.** EDX elemental composition maps of oxygen (green), phosphorus (red) and nitrogen (blue) of IHMA87 wild-type and Δ*purD* in plasma (left). EDX spectra (right) of highlighted locations (red = granules, cyan = cytoplasm), major peaks are assigned. Data show elevated levels of phosphorus and oxygen in the granule, indicating that the identified objects are polyP.

spectroscopy (EDX) analysis (Fig 5B). The formation of polyP granules upon incubation in plasma was also detected in the two BSI isolates, PaG1 and PaG7, showing a conserved response to plasma (S4 Fig). Formation of polyP granules was readily observed in bacteria incubated with HIP, suggesting the formation of polyP granules is triggered in response to the nutritive conditions of the plasma itself rather than active CS components (S5 Fig).

PolyP are synthesized by polyphosphate kinases (Ppk) [58,61], encoded by three *ppk* genes in *P. aeruginosa*. We hypothesized that the polyP could help *P. aeruginosa* to cope with plasma-related stress, and we therefore anticipate a deleterious effect of Ppk1 inactivation on IHMA87 survival in plasma. First, we interrogated the Tn-seq datasets to evaluate the relevance of polyP in evaders' formation. Indeed, sequencing profiles highlighted that transposon mutants in the three Ppk-encoding genes were less represented in plasma (*e.g.* $Log_2(FC)_{ppk1}$ = -2.33, Fig 6A and S1 Table). To confirm those data, we constructed mutants carrying deletions in *ppk1* (*IHMA87_05726*) and *ppk2* (*IHMA87_00143*) genes, and a double *ppk1-ppk2* mutant. While simple mutants still displayed some polyP granules upon incubation in plasma, the double mutant was devoid of granules (Fig 6B). In agreement with Tn-seq datasets, plasma killing assays showed that individual mutants displayed increased sensitivity to plasma compared to the wild type, while the double *ppk1-ppk2* mutant was almost completely eliminated (Fig 6C). Together, those results confirm that polyP granule formation is an important factor allowing *P. aeruginosa* to persist in the human plasma.

Altogether, incubation of *P. aeruginosa* in human plasma triggers the formation of electron dense polyP granules. In the plasma-tolerant strain, Δ*purD*, the granules appeared bigger than in the parental strain, probably mimicking the purine-depletion stress response. PolyP biosynthetic genes, *ppk1* and *ppk2*, encoding polyphosphate kinases, were critical for persistence in plasma. Thus, the adaptation of bacteria to plasma through formation of energy storage granules may favor the selection of evaders that resist MAC killing.

## Specificity of mutant resilience toward MAC-induced killing

Finally, to examine the possible mechanisms deployed by selected mutants to resist the bactericidal activity of the MAC in plasma, we set up experiments to monitor several steps in the complement cascade. Complement activation by bacteria–both the classical and the alternative pathways–was examined by measuring the residual complement activity toward sensitized erythrocytes as described in Dumestre-Pérard et al., 2008 [62]. After incubation with the different mutants, all plasma samples had the same residual complement activity as plasma incubated with the parental strain (Fig 7A and 7B), suggesting that the mutants did not alter complement activation.

To investigate whether the mutants prevent the formation of C3b and MAC (C5b-9) complexes, we first assessed deposits of CS components on the bacterial surface by flow cytometry (Fig 7C–7H). As shown in Fig 7C–7D, increased C3b deposition was observed at 15 min for the Δ*bioB* mutant. When bacteria were exposed to pooled human serum supplemented with fluorescently-labeled C9 (Fig 7E and 7F), similar levels of C9 deposition were observed for most strains, with an apparent increase for the Tn::$P_{srg}$ mutant.

Finally, we used flow cytometry with an anti-C5b-9 antibody to investigate MAC polymerization (Fig 7G and 7H). No significant variation in the amount of C5b-9 complex was detected between the different mutants suggesting that, overall, MAC coating was unaltered in all the mutants.

We therefore assessed the functionality of the MAC pores inserted in the outer membrane by comparing survival of the different bacterial strains in serum in the presence of nisin. The antimicrobial molecule nisin normally is inactive against Gram-negative bacteria with an

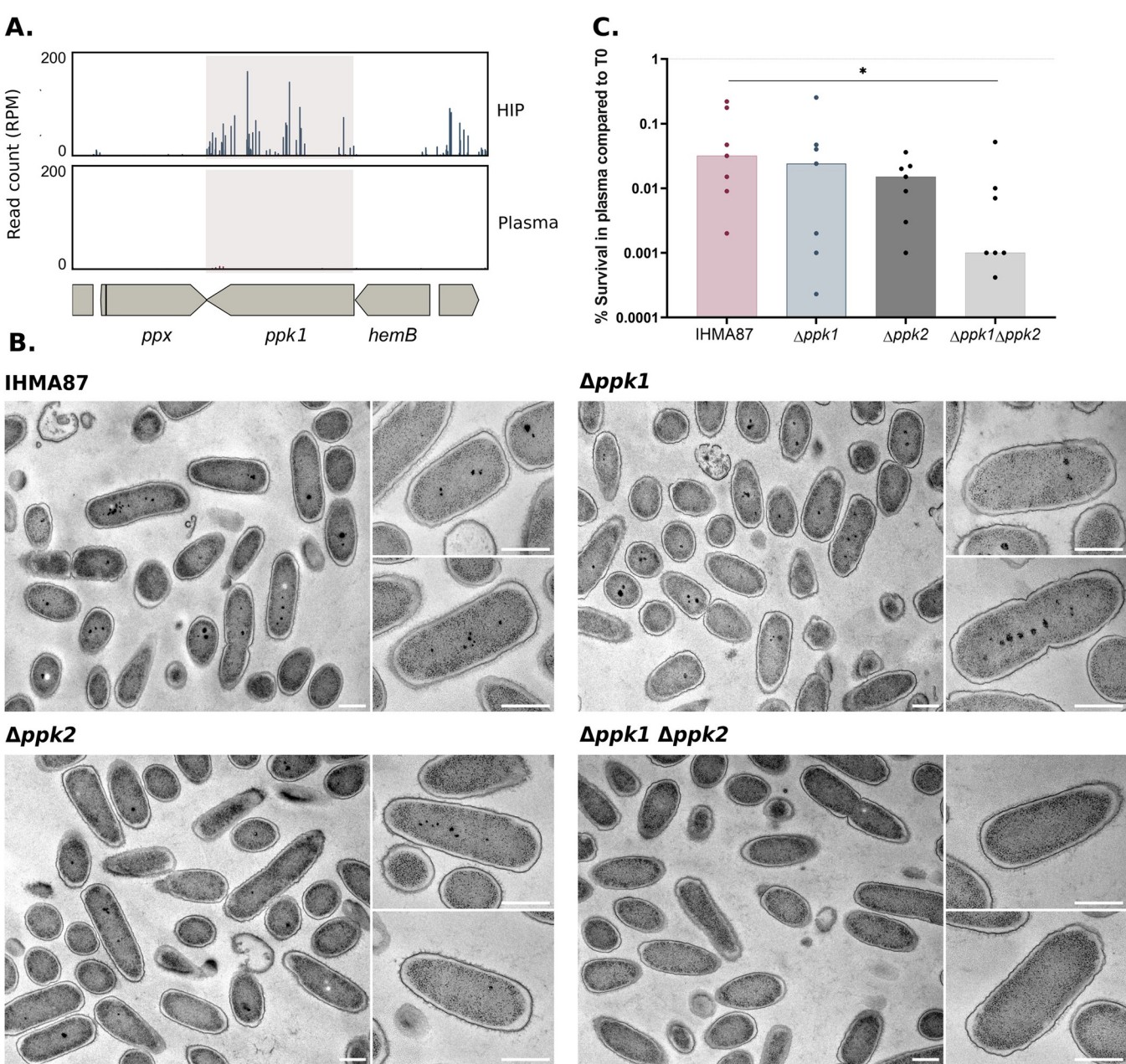

**Fig 6. Polyphosphates are critical for evaders' survival in plasma. A.** Zoom in on Tn-seq profiles of *ppk1* gene with 2000 bp upstream and downstream, showing normalized numbers of reads in input (HIP) and output (plasma) samples. **B.** Transmission electron microscopy images of IHMA87, Δ*ppk1*, Δ*ppk2* or Δ*ppk1*Δ*ppk2* after 1h-incubation in human plasma. Scale bar = 500 nm. **C.** Survival of IHMA87 wild-type strain, individual Δ*ppk1*, Δ*ppk2* or double Δ*ppk1*Δ*ppk2* deletion mutants following incubation for 3 h in plasma. Survival rates were calculated based on CFU measurements and the median of all independent experiments is represented by the histogram. Statistical analysis was performed and *p*-value <0.05 is indicated with '*'.

intact outer membrane [17]. However, MAC-induced membrane damage allows the compound to enter the bacterial cells, resulting in killing. As previously shown for plasma, wild-type *P. aeruginosa* IHMA87 is highly sensitive to killing in serum, with very efficient bacterial elimination (Fig 7I). In contrast, increased survival was recorded for all mutants. Interestingly, the presence of nisin led to the death of most of the evader population. Similarly, numbers of the Tn::$P_{srg}$ mutant were reduced in the presence of nisin, indicating that MAC-induced outer

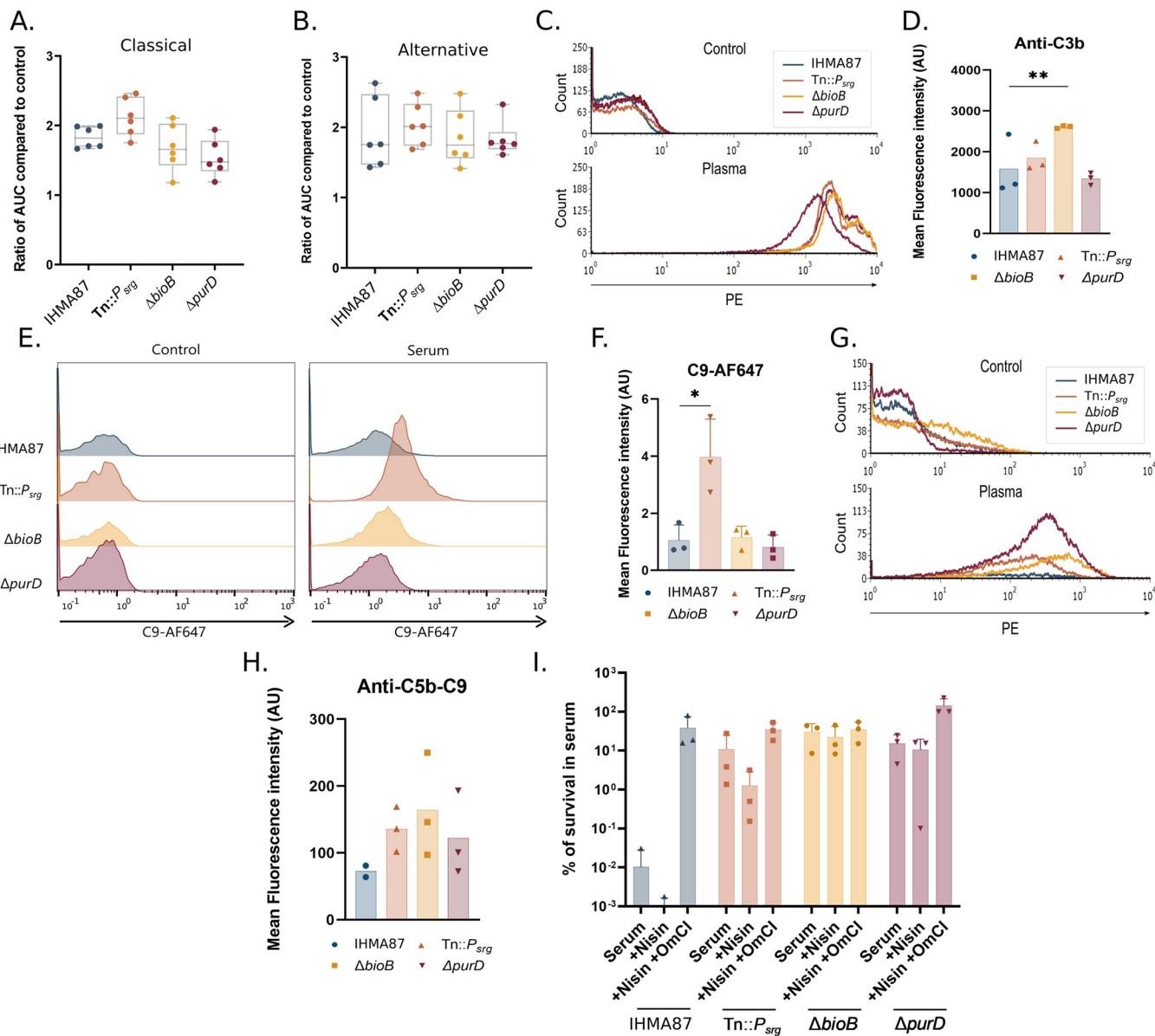

**Fig 7. Mutants' response to MAC-dependent killing.** Residual complement activity of the classical (**A.**) and alternative (**B.**) complement pathways was measured after bacterial challenge in plasma, using lysis kinetics of erythrocytes, as described in Materials and Methods. Area under the curve (AUC) values were determined for each series and expressed relative to the 90% plasma pool control. n = 3. **C.** C3b deposition on bacterial surface after incubation for 15 min in 0% (Control) or 90% plasma. Deposition was determined by FACS using the C3b specific PE-labeled antibody. **D.** Mean fluorescence intensities of the plasma condition presented in **C.** n = 3. **\*\*** $p$-value $< 0.001$. **E.** C9-AF647 insertion into bacterial membrane after 30 min in 0% (Control) or 3% pooled human serum and **F.** mean fluorescence intensities (**\*** $p$-value $< 0.05$). **G.** C5b-9 deposition on bacterial surface after 15 min incubation in 0% (Control) and 90% plasma. Deposition was determined by FACS; mean fluorescence intensities are plotted (**H.**). n = 3. **I.** Bacterial survival in pooled human serum or in the presence of nisin with or without the complement inhibitor OmCI.

membrane damage does occur in this strain. These observations suggest that outer membrane damage by the MAC itself was not sufficient to kill this mutant or the evader population. In contrast, survival of the Δ*bioB* and Δ*purD* strains was unaltered by the addition of nisin to serum. Survival of all strains could be rescued by addition of C5 inhibitor OmCI, confirming that the killing is complement-mediated. Taken together, these data suggest that the Δ*bioB*, Δ*purD*, and Tn::P*srg* mutants survive better than the wild-type strain in serum. The Tn::P*srg*

mutant interferes at a different stage of MAC-dependent killing than the Δ*bioB* and Δ*purD* mutants.

## Discussion

The aim of this study was to explore factors used by *P. aeruginosa* to persist in human plasma. *P. aeruginosa* displays strain-dependent and highly variable survival in human plasma, ranging from fully resistant to sensitive. The majority of strains that are sensitive at the whole population level form a persistent sub-population with yet uncharacterized features [11]. Through this study, we identified polyP, biotin, and a small predicted periplasmic protein (SrgA, 9kDa) as novel determinants of *P. aeruginosa* resilience to the human CS. Other groups have performed genome-wide screens in complement-resistant strains of a number of pathogens in human serum [51,63–66]. Compared to those studies, we confirmed that long OSA (*wzz1*) and exopolysaccharides (both alginates and Psl) are important for the survival of *P. aeruginosa* in human plasma. Similarly, in agreement with genetic screens that identified determinants of bacterial persistence following antibiotic treatment [67,68], our results show that *P. aeruginosa* persistence in plasma is a multifactorial and probably stochastic phenomenon (Fig 8).

In our search to identify the mechanisms used by the mutants to escape complement-mediated killing, we found that Tn::*P_srg*, Δ*bioB*, and Δ*purD* mutants do not impede complement

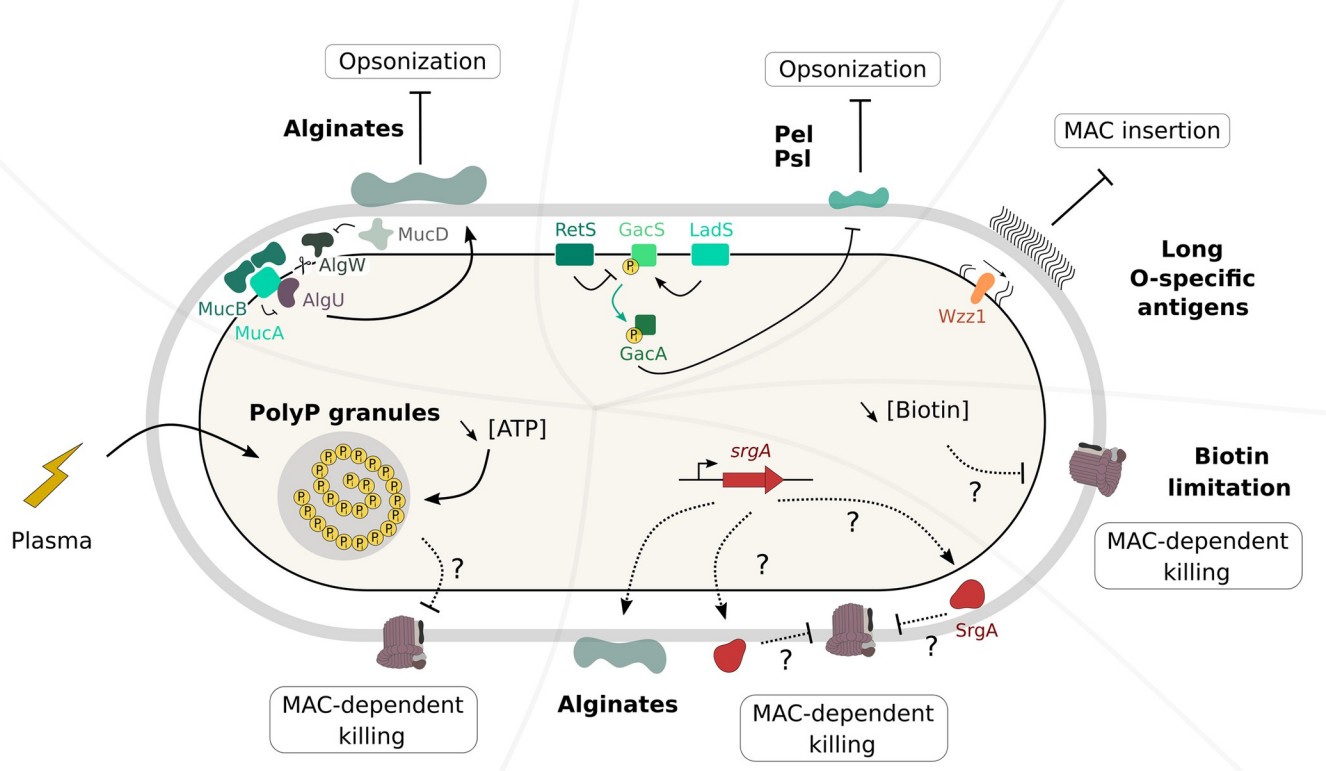

**Fig 8. Complement evasion by *P. aeruginosa* is a multifactorial phenomenon.** In addition to described complement-mediated killing evasion mechanisms including exopolysaccharides (alginates, Psl and Pel) and long O-specific antigen, which limit bacterial opsonization or MAC insertion, we herein describe three novel evasion determinants. Decreased biotin production and SrgA overexpression, increase *P. aeruginosa* persistence and tolerance, respectively. Finally, bacterial incubation in plasma leads to the production of polyP granules, which are critical for the formation of evaders. The mutants in *ppk* genes, unable to produce polyP, have a significant decrease in the number of evaders formed upon plasma challenge. The three mechanisms described here impact bacterial survival in plasma by impeding MAC lytic activity.

activation. Rather, they probably modulate the function of MAC itself. MAC is an 18-unit oligomeric multiprotein pore formed by sequential addition of C5b, C6, C7, C8, and finally C9, to create a ring in the bacterial outer membrane [69]. Although the composition of MAC has been known for a number of years, its contribution to bacterial cell death and the mechanism causing bacterial death following MAC insertion remain debated [70]. Recent reports suggest that complement-resistant bacteria can specifically block C9 polymerization [71]. SrgA–7.1-kDa after cleavage of its signal peptide–may directly interfere with MAC assembly. Alternatively, SrgA may act on the peptidoglycan or other components of the Gram-negative envelope to impede MAC function. Despite several attempts, we have so far been unable to detect SrgA using protein tags in *P. aeruginosa*, but preliminary visualization of the protein in *E. coli* confirms that the *srgA* gene product is of a peptide nature. In contrast, the nature of the *srgB* and *srgC* gene products is speculative.

The Δ*bioB* and Δ*purD* mutants, as well as Tn::*P*$_{srg}$, to a lower extent, displayed resistance to the outer-membrane-impermeable polycyclic antibacterial peptide nisin (Fig 7I). Thus, in these mutants, the MAC pore was insufficient to allow nisin uptake through the permeabilized outer membrane. This difference in sensitivity could be due to defective pore insertion or to more resistant inner membranes. In contrast, in evaders that arise from the wild-type genetic background, the outer membrane seems to have been effectively damaged by the MAC, suggesting that evaders have a more resistant envelope or can repair envelope damage more rapidly. Further characterization of evader biology and comparison to the bulk population will be needed to identify and further explore the molecular mechanisms leading to the emergence of evaders.

One highlight of our work is that polyP are critical in evaders' formation. Upon incubation in active or inactivated plasma, wild-type IHMA87 bacteria as well as BSI isolates form polyP granules that are energy storage structures [58,60], and Δ*purD* mutant with low ATP levels displayed even bigger granules. Moreover, Ppk enzymes responsible for polyP synthesis are found critical for evaders, clearly suggesting that bacteria sense and respond to the stressful plasma environment (Fig 6).

The number of plasma evaders present following incubation in plasma varies between experiments, strains and the plasma used ([11] and this work). Therefore, it is possible that the plasma itself, as a stressful environment for bacteria, stochastically triggers the emergence of evaders. One stress response in *P. aeruginosa* is the formation of polyP granules that were observed in several strains upon incubation in plasma. PolyP are involved in many virulence phenotypes including motility and biofilm formation [59], and *ppk* mutants in *Neisseria* species were found previously to be strikingly more sensitive to killing by serum [72,73].

PolyP may impair MAC activity *per se* given the fact that plasma treated Δ*purD* population, which produces more polyP, is resistant to nisin treatment (Fig 7I). The mechanism by which this occurs is still unclear. PolyP could directly inhibit complement activity by inhibiting C5b-6 complex formation as proposed by Conway *et al*., 2019 [74], however, bacterial polyP would need to be exported to the periplasm. PolyP could also modify outer membrane constituents, as proposed in *Campylobacter jejuni*, leading to a reduced MAC insertion or to increased envelope resistance to killing [75].

Given the multifaceted evasion to plasma-dependent killing, both the genetic and phenotypic diversity of *P. aeruginosa* strains [76,77], and their variable levels of survival in human plasma [11], we expect that both common and strain-specific determinants counteracting MAC-dependent killing will be discovered. Only by screening several representative bacterial strains in parallel can we hope to obtain an overall picture of the *P. aeruginosa* plasma and blood resistome/persistome.

In an era where bacterial resistance to antibiotics has become a global health concern, it is highly relevant to explore any alternative means to eliminate pathogens. Harnessing the CS or

MAC may be one possibility. Recent studies have proposed to therapeutically target Ppk enzymes, which do not have any homologue in humans, to dampen *P. aeruginosa* virulence [78,79]. Such treatment could result in elimination of evaders from plasma and help avoid treatment failure. For example, strategies to enhance MAC formation could be combined with classical antibiotics, phages, or antibodies [80–82]. However, our current comprehension of how various pathogens escape complement-mediated killing remains very limited. Better understanding of the molecular mechanisms involved in pathogen-complement interplay should help to design strategies to combat multidrug resistant bacteria.

## Materials and methods

### Bacterial strains and genetic manipulations

Bacterial strains, plasmids, and primers are listed in S3 and S4 Tables. *E. coli* and *P. aeruginosa* were grown in LB at 37°C with shaking (300 rpm). *P. aeruginosa* was selected on LB plates containing 25 μg/mL irgasan. Antibiotic concentrations were as follows: 75 μg/mL gentamicin, 75 μg/mL tetracycline, and 300 μg/mL carbenicillin for *P. aeruginosa*; and 50 μg/mL gentamicin, 10 μg/mL tetracycline, and 100 μg/mL ampicillin for *E. coli*. To create deletion mutants, upstream and downstream flanking regions (approximately 500 bp) were amplified from genomic DNA by PCR using appropriate primer pairs (sF1/sR1, sF2/sR2). Overlapping fragments were cloned into *Sma*I-digested pEXG2, pEX100T, or pEX18Tc by Sequence- and Ligation-Independent Cloning (SLIC—[83]). Plasmids were used to transform *E. coli* Top10 competent cells, and their sequences were verified (Eurofins). Allelic exchange vectors derived from pEXG2, pEX100T, or pEX18Tc were introduced into *P. aeruginosa* by triparental mating, using pRK600 as a helper plasmid. Merodiploids, resulting from homologous recombination, were selected on LB agar plates containing irgasan and the appropriate antibiotic. Single merodiploid colonies were streaked on NaCl-free LB plates with 10% sucrose (w/v) to select for plasmid loss. Resulting sucrose-resistant clones were screened for antibiotic sensitivity and gene deletion by PCR.

### Generation of the IHMA87 transposon library

The IHMA87 transposon mutant library was constructed essentially as previously described [41,84]. Briefly, IHMA87, grown overnight at 42°C under shaking, was mixed with two *E. coli* strains carrying pRK2013 [85] and pBTK24. The conjugation mixtures–containing 100 μL of each bacterial culture adjusted to an optical density at 600 nm ($OD_{600}$) of 1 –were centrifuged, washed with LB, deposited on pre-warmed LB plates and incubated at 37°C for 5 h. Approximately 100 conjugation mixtures yielded a library of > 300,000 mutants. Bacteria were scraped off into liquid LB, and aliquots were plated on LB plates containing irgasan (25 μg/mL) and gentamicin (75 μg/mL). Following growth at 37°C, colonies were collected directly into LB containing 20% glycerol, aliquoted and stored at -80°C.

### Preparation of pooled plasma

Pooled plasma was prepared from heparinized blood of human healthy donors provided by the French National Blood Institute (EFS, Grenoble, France). Fresh blood was centrifuged for 10 min at 1,000 x g at room temperature. Supernatants from ten distinct donors were pooled, filtered through a 0.45-μm membrane and aliquoted prior to storage at -80°C until needed. The same procedure was applied to citrated blood from 30 healthy donors. Before use, pooled plasma aliquots were thawed on ice, centrifuged for 10 min at 10,000 rpm and filtered through a 0.22-μm membrane. For the heat inactivated plasma (HIP) condition, plasma was thawed,

and heat inactivated for 30 min at 56˚C, centrifuged for 10 min at 10,000 rpm, and filtered through a 0.22-μm membrane.

## Plasma screening of *P. aeruginosa* library

A vial containing the IHMA87 mutant library was thawed on ice, transferred to a flask containing 30 mL of LB and grown for 16 h under agitation at 27˚C, to limit bacterial growth. The next day, the bacterial suspension was diluted to $OD_{600nm}$ ~0.1 and grown at 37˚C under agitation. Meanwhile, plasma and HIP were prepared. When cultures had reached an $OD_{600nm}$ ~1, bacteria were harvested and resuspended in PBS supplemented with calcium and magnesium ions (Thermofisher Scientific) before exposing them to plasma, HIP, or LB at a concentration equivalent to $2.25x10^7$ CFU/mL (90% final plasma concentration). The precise initial CFU count for each experiment was determined by serial dilution. Samples were then incubated at 37˚C for 3 h on a rotating wheel. At the end of the challenge, an aliquot of each sample was transferred to LB and grown overnight; another aliquot was spread on LB agar plates to isolate individual mutants. Finally, bacterial survival rates were calculated based on the CFU counts for each sample. The same protocol was applied with the wild-type strain in parallel to compare survival rates. The next day, culture aliquots were harvested and stored for further mutant isolation. DNA was extracted from about $1x10^9$ bacteria for each sample.

## Illumina library construction, sequencing, and data analysis

**Genomic DNA.** Bacterial pellets from the screen in either HIP (Input) or plasma (Output) were used to prepare gDNA. A classical DNA-preparation protocol was applied, involving an SDS-NaCl lysis step (2% SDS, 0.15 M NaCl, 0.1 M EDTA pH8, 0.6 M Sodium perchlorate) followed by two successive phenol-chloroform extractions. The gDNA was precipitated with 100% cold ethanol and diluted in 200 μL TE buffer (10 mM Tris-HCl, 1 mM EDTA, pH8), producing a final concentration of 50–100 ng/μL. DNA was mechanically sheared at 4˚C with a Qsonica sonicator (Q700) in 15-s pulses for a total of 20 min. DNA fragments were then concentrated on columns (Monarch PCR & DNA Cleanup Kit, NEB). The size of the DNA fragments generated (150–400 bp) was verified on 2% agarose gel.

**Library construction.** Fragmented gDNA (1 μg) was end-repaired using an End-repair module (NEB#E6050), and dA-tailed using Klenow (NEB#6053). Short and Long adaptors (S4 Table) were annealed in 10 mM $MgCl_2$ in a thermocycler programmed to decrease the temperature by 1˚C/cycle between 95˚C and 20˚C. gDNA fragments were ligated with annealed adaptors overnight at 16˚C in the presence of T4 DNA ligase (NEB#M0202). Bands from 200 to 400 bp were selected by extraction from a 2% agarose gel using a Monarch DNA Gel Extraction Kit (NEB#T1020L). Purified adaptor-DNA fragments were amplified by Phusion polymerase (NEB#M0530) in combination with specific primers (PCR1 Tn-spe direct and PCR1 Adaptor comp.). A second amplification round was performed with a primer bearing a P5 Illumina sequence and P7 indexed Illumina primers (NEB#E7335S). Primer sequences are available on request. After each step, products were quantified using a Qubit dsDNA HS Assay kit (Q33230). The quality of the libraries was assessed on an Agilent Bioanalyzer 2100 using high-sensitivity DNA chips (Ref #5067–4626). Constructs were sequenced on an Illumina NextSeq High (I2BC, Saclay Paris).

**Data analysis.** Sequencing reads were trimmed and aligned with the IHMA87 genome using Bowtie2 [86]. Htseq-count [87] was then used to determine read-counts for each feature. DESeq2 [88] was applied to determine differential representation of insertion mutants between the plasma and HIP samples. To analyze insertions in intergenic regions, an annotation file was generated where intergenic regions were attributed to the upstream gene regardless of its

orientation. The analysis protocol described above, using Htseq-count and DESeq2, was applied. Tn-seq was performed on biological duplicates.

## Plasma killing assay

Killing assays were performed as described in Pont et al., 2020 [11]. Unless stated otherwise, bacterial pre-cultures were diluted in LB to $OD_{600nm}$ = 0.1 and grown at 37˚C under agitation until $OD_{600nm}$ ~ 1. Bacteria were resuspended in PBS +/+ (Thermofisher Scientific) and incubated in plasma at a concentration of $2.25 \times 10^7$ CFU/mL (90% final plasma concentration). The precise initial CFU count used in each experiment was determined by serial dilution. Bacterial survival rates were calculated based on CFU counts at the indicated time(s) and compared to the initial count. ATP, biotin, and purine were added to the plasma before adding bacteria.

## Transcriptomics

**RNA extraction.** Bacterial cultures were diluted to an $OD_{600nm}$ of 0.1 and grown at 37˚C under agitation until they reached an $OD_{600nm}$ of 1. RNA was isolated using the hot phenol-chloroform method. Briefly, bacteria were lysed in a hot (60˚C) phenol solution (2 mM EDTA, pH8, 1% SDS, 40 mM sodium acetate in acid phenol (Invitrogen #15594–047)). RNA was then isolated by successive phenol-chloroform extractions, followed by a cold chloroform extraction, and ethanol precipitation. Any residual genomic DNA was eliminated by Turbo DNase (Invitrogen #AM1907) treatment. Quantity and quality of total RNA were assessed using an Agilent Bioanalyzer.

**Construction of RNA-seq libraries.** Following RNA extraction, ribosomal RNAs (rRNAs) were depleted using a ribominus kit (Invitrogen) according to the manufacturer's instructions. Depletion quality was verified on an Agilent Bioanalyzer using RNA pico chips before concentrating RNA by ethanol precipitation.

The NEBNext Ultra II directional RNA library prep kit for Illumina (NEB) was used to build cDNA libraries in preparation for sequencing, starting with 50 ng of rRNA-depleted RNA. The manufacturer's instructions were followed. The quality and concentration of cDNA libraries were assessed using Agilent Bioanalyzer High-sensitivity DNA chips.

**Sequencing and data analysis.** DNA was sequenced at the Institute for Integrative Biology of the Cell (I2BC http://www.i2bc.paris-saclay.fr) high-throughput sequencing facility using an Illumina NextSeq500 instrument. More than 12 million 75-bp single-end reads were obtained per sample. Processed reads were mapped onto the IHMA87 genome (available on NCBI CP041354 and CP041355 accession numbers, for the chromosome and plasmid sequences, respectively [41] using Bowtie2 [86] and the readcount per feature was calculated using Ht-seq count [87]. Finally, DESeq2 [88] was applied to determine differential gene expression between the Tn::$P_{srg}$ and wild-type strains. RNA-seq experiments were performed on biological duplicates.

## Alginate assay

Equivalent amounts of $OD_{600nm}$-adjusted bacterial cultures were spread on *Pseudomonas* isolation agar (PIA) plates and incubated for 24 h at 37˚C. Bacterial lawns were flushed with saline solution (0.9% NaCl) and scraped off the plates. Bacteria were pelleted, and supernatant was stored in a separate tube. The bacterial pellet was resuspended in LB and incubated on a rotating wheel for 1–2 h to disrupt bacterial aggregates. The titer of the suspension was determined by serial dilution and CFU counting. Alginate was assayed as described in Knutson and Jeanes, 1968 [89] and May and Chakrabarty, 1994 [90]. Briefly, 600 μL of ice-cold borate

working solution (borate stock solution (4 M $BO_3{}^{3-}$) diluted to a final concentration of 100 mM in $H_2SO_4$) was added to 70 μL of the bacterial supernatant, saline solution for a blank, and to commercial sodium alginate solution to produce a standard curve. On ice, 20 μL of carbazole solution (0.1% carbazole (Sigma #442506) in 100% ethanol) was added, and samples were incubated at 55°C for 30 min after mixing. Finally, absorbance was measured at 530 nm.

## Determination of intracellular bacterial ATP concentration

To measure ATP in bacteria grown in LB, about $1 \times 10^8$ bacteria were pelleted and resuspended in PBS. HIP was inoculated with a final concentration of $2.025 \times 10^7$ bacteria/mL in PBS (90% final HIP concentration). Samples were incubated for 2 h at 37°C on a rotating wheel, centrifuged and resuspended in PBS. Bacterial aggregates were gently dissociated by sonication using a Qsonica sonicator (Q700) for 45 sec at 10% intensity in 5-s-ON-5-s-OFF intervals. For the two conditions, 10-fold serial dilutions were prepared in 96-well plates, and precise bacterial counts were determined based on CFUs. The ATP concentration was determined using the BacTiter-Glo kit (Promega, #G8230). Briefly, equal volumes of bacterial suspension and BacTiter-Glo reagent were mixed in white 96-well Lumitrac plates (Bio-Rad #655075) and incubated for 5 min under shaking. A standard curve (0.1 to 100 nM ATP) was established for each experiment. Luminescence was measured on a plate reader (Spark, TECAN). The ATP concentration was then expressed relative to the CFU count. Experiments were performed in more than three biological replicates.

## Residual complement activity

**Sample preparation.** Bacteria grown to $0D_{600nm} \sim 1$ were resuspended in PBS +/+ (Thermofisher Scientific) and incubated in plasma at a concentration of $2.25 \times 10^9$ CFU/mL (90% final plasma concentration), or PBS alone as control. Samples were incubated for 3 h at 37°C on a rotating wheel to trigger complement activation by bacteria. The initial CFU count was determined by serial dilution. At the end of the incubation, samples were centrifuged, and supernatant was filtered through a 0.22-μm membrane to remove any remaining bacteria. Samples were aliquoted and frozen at -80°C until use. Residual complement activity in the bacterial/plasma supernatant was measured using a total hemolytic assay to study classical and alternative pathways, as described in Dumestre-Pérard et al., 2008 [62].

**Classical complement hemolytic activity.** Briefly, sheep erythrocytes (Orgentec) were sensitized using anti-erythrocyte antibodies (1/40 000; Hemolysin, Orgentec) for 15 min at 37°C. Bacterial/plasma supernatant (25 μL) was mixed with 3 mL of sensitized erythrocytes ($4 \times 10^6$ cells/mL) diluted in $DGVB^{2+}$ (2.5% glucose, 0.05% gelatin, 2.5 mM Veronal, 72.5 mM NaCl, 0.15 mM $Ca^{2+}$, 0.5 mM $Mg^{2+}$), and lysis kinetics were monitored based on $OD_{660nm}$ using a spectrophotometer (Safas UVmc2). Area under the curve values were determined for each series and expressed relative to the 90% plasma pool control.

**Alternative complement hemolytic activity.** Briefly, rabbit erythrocytes (Ecole Nationale Vétérinaire de Toulouse) were washed three times using DGVB-Mg-EGTA (2.5% glucose, 0.05% gelatin, 2.5 mM Veronal, 72.5 mM NaCl, 5 mM $Mg^{2+}$, 3 mM EGTA). Bacterial/plasma supernatant (150 μL) was added to 3 mL of a rabbit erythrocyte suspension (diluted in DGVB-Mg-EGTA at 1/250) at 37°C, and lysis kinetics were monitored as described for classical complement activity.

## C3b and C5b-9 binding assays

Bacteria ($4 \times 10^7$) were incubated for 15 min in a 90% heparinized plasma pool at 37°C. Incubation was stopped by diluting the bacterial suspension in ten volumes of PBS containing 20

mM EDTA (pH8). C3b binding was detected with PE-conjugated mouse monoclonal anti-C3b/iC3b (5 μg/mL, BioLegend #846103) and C5b-9 membrane deposition/insertion was detected with mouse monoclonal anti-C5b-9 antibody (8 μg/mL, Abcam #ab66768). Antibody binding was revealed with PE-conjugated goat anti-mouse antibodies (2.5 μg/mL, Abcam #ab97041). The negative control consisted of bacteria in 0% plasma (100% PBS). Samples were analyzed on a FACSCalibur (Becton Dickinson); data were acquired and analyzed using FCSExpress6 Software.

### C9 binding

*P. aeruginosa* strains expressing green fluorescent protein (GFP) under the control of the constitutive $P_{X2}$ promoter were grown overnight in LB supplemented tetracycline at 37˚C, with shaking. Bacteria were subcultured 1:30 in LB supplemented with the antibiotic and allowed to grow to a mid-log $OD_{600nm}$ of 0.5 at 37˚C, with shaking. Bacteria were pelleted by centrifugation, washed, and resuspended at a final $OD_{600nm}$ of 0.05 in RPMI + 0.05% human serum albumin (HSA) for the experiments. Bacteria were then exposed to pooled human serum supplemented with 100 nM C9-AF647 and incubated for 30 min at 37˚C, with shaking. To count CFU, bacteria were serially diluted in MilliQ water, spotted on LB plates, and incubated overnight at 37˚C. For flow cytometry, bacteria were diluted 10-fold in RPMI-HSA and analyzed by flow cytometry (MACSquant VYB; Miltenyi Biotech). Bacteria were gated on GFP-expression and subsequently analyzed for AF647 fluorescence. Flow cytometry data was analyzed using FlowJo V.10.

### Nisin assay

Bacteria were grown overnight in LB at 37˚C, with shaking, subcultured 1:30 in LB to a mid-log $OD_{600nm}$ of 0.5 at 37˚C, with shaking, pelleted by centrifugation, washed, and resuspended at a final $OD_{600nm}$ of 0.05 in RPMI + 0.05% HSA for the experiments. Bacteria were then exposed to human pooled plasma supplemented with buffer, 3 μg/mL nisin, or 3 μg/mL nisin and 10 μg/mL OmCI and incubated for 3 h at 37˚C. Serial dilutions in MilliQ water were spotted on LB plates for incubation overnight at 37˚C to determine CFU the next day.

### Transmission electron microscopy

Bacterial pre-cultures were diluted to an $OD_{600nm}$ of 0.1 in flasks containing 30 mL of LB and incubated at 37˚C, with shaking until $OD_{600nm}$ of 1. Plasma was then inoculated with bacterial suspension to reach a bacterial concentration of $1.125 \times 10^9$ bacteria/mL and incubated on a rotating wheel for 1 h at 37˚C. At the end of the challenge, bacteria were pelleted by centrifugation and resuspended in LB. In parallel, bacteria grown in LB to an $OD_{600nm}$ of 1 were pelleted by centrifugation and resuspended in LB. Samples were treated as described in Mohamed et al., 2021 [91]. Briefly, the cell pellet was spread on the 200-μm side of a 3-mm type A gold plate (Leica Microsystems) covered with the flat side of a 3-mm type B aluminum plate (Leica Microsystems) and vitrified by high-pressure freezing in a HPM100 system (Leica Microsystems). After freezing at -90˚C in acetone with 1% $OsO_4$, samples were slowly warmed to -60˚C before storing for 12 h. Then, the temperature was increased to -30˚C and the samples were stored for a further 12 h. Samples were then warmed to 0˚C, incubated for 1 h, and cooled once again to -30˚C (AFS2; Leica Microsystems). Vitrified samples were rinsed four times in pure acetone before infiltration with progressively increasing concentrations of resin (Epoxy embedding medium, Sigma) in acetone while increasing the temperature to 20˚C. Pure resin was added at room temperature. After polymerization, thin (70-nm) sections were prepared using a UC7 ultramicrotome (Leica Microsystems) and collected on 100-mesh copper grids

coated with Formvar carbon. The thin sections were post-stained for 5 min with 2% uranyl acetate, rinsed with water and incubated for 2 min with lead citrate. Samples were observed using a Tecnai G2 Spirit BioTwin microscope (FEI) operating at 120 kV with an Orius SC1000B CCD camera (Gatan).

## Granules analysis

For the chemical analysis of granules, 200 nm-thin sections, collected on carbon/formvar coated copper grids, were observed in scanning/transmission electron microscopy (STEM) mode and analyzed by STEM (HAADF) detector, on a Tecnai OSIRIS microscope (FEI) operated at 200 kV. The chemical composition of regions of interest was then analyzed by EDS using the ESPRIT 1.9 software (Bruker).

## Bioinformatics analysis and protein conservation

Protein sequence files from all *P. aeruginosa* strains with a closed assembled genome were retrieved from the Pseudomonas genome database, and *srg* conservation was investigated using the Blast Reciprocal Best Hits program [92], selecting for 90% coverage and at least 50% identity. Signal peptides were predicted on the SignalP website [93].

## Statistical analysis

Sigmaplot software was used to perform statistical analysis. Multiple-group comparisons were performed by applying one-way ANOVA or Kruskal-Wallis tests, depending on whether data was normally distributed. Subsequently, a Student-Newman-Keuls pairwise comparison was applied. In some cases (as indicated), non-normally distributed data were converted to normally distributed datasets by $\log_{10}$-transformation. Figures were designed using GraphPad Prism9.

## Supporting information

**S1 Fig. Overexpression of *srg* operon leads to a population tolerant to plasma killing.** Survival kinetics of IHMA87 wild-type strain (same data as presented in Fig 2B) and of Tn::$P_{srg}$ in plasma over 6h incubation measured by CFU counting (n = 5).
(DOCX)

**S2 Fig. Depletion in biotin and purine increase bacterial survival to plasma independently from one another. A.** The survival of IHMA87 WT, Δ*bioB*, Δ*purD* and Δ*purD*Δ*bioB* was estimated by CFU counting and the median of all independent experiments is represented by the histogram. Statistical analysis was performed and p-value <0.05 or 0.01 are indicated with '*' and '**', respectively. **B.** Transmission electron microscopy images of IHMA87Δ*purD*Δ*bioB* after 1h-incubation in plasma. Scale bar = 500 nm.
(DOCX)

**S3 Fig. ATP concentration in LB and plasma pools.** Measurement of ATP levels in different plasma pools. ATP was not detected in LB. n = 3.
(DOCX)

**S4 Fig. Incubation in plasma triggers polyphosphate granules formation. A.** Transmission electron microscopy images of Δ*bioB* after growth in LB (left) or 1h-incubation in human plasma (right). Arrows show small-sized granules. **B.** Transmission electron microscopy images of two BSI isolates PaG1 and PaG7 after growth in LB (left) or 1h-incubation in human

plasma (right). Scale bar = 500 nm.
(DOCX)

**S5 Fig. Heat inactivated plasma triggers the formation of polyP granules.** Transmission electron microscopy images of IHMA87 after 1h-incubation in heat-inactivated plasma (HIP, left) or in native human plasma (right). Scale bar = 500 nm.
(DOCX)

**S1 Table. Tn-seq data.**
(XLSX)

**S2 Table. RNA-seq differential gene expression data.**
(XLSX)

**S3 Table. Bacterial strains and plasmids.**
(DOCX)

**S4 Table. Oligonucleotides used for PCRs and Tn-seq.**
(DOCX)

## Acknowledgments

We thank the International Health Management Association, USA, for making available the *P. aeruginosa* IHMA87 strain. The authors thank Emily Rey for her help with mutant analyses. M.JM thanks Twitter followers for indicating literature on PolyP granules. The sequencing experiments benefited from the facilities and expertise of the high throughput sequencing core facility at I2BC (Centre de Recherche de Gif– http://www.i2bc.paris-saclay.fr/). The IBS Electron Microscope facility is supported by the Auvergne Rhône-Alpes Region, the Fonds Feder, the Fondation pour la Recherche Médicale, and GIS-IBiSA.

## Author Contributions

**Conceptualization:** Manon Janet-Maitre, Stéphane Pont, Chantal Dumestre-Perard, Bart W. Bardoel, Suzan H. M. Rooijakkers, François Cretin, Ina Attrée.

**Data curation:** Manon Janet-Maitre, Julian Trouillon, Mylène Robert-Genthon.

**Formal analysis:** Manon Janet-Maitre, Frerich M. Masson, Sylvie Elsen, Bart W. Bardoel, Suzan H. M. Rooijakkers, François Cretin, Ina Attrée.

**Funding acquisition:** Suzan H. M. Rooijakkers, Ina Attrée.

**Investigation:** Manon Janet-Maitre, Stéphane Pont, Frerich M. Masson, Serena Sleiman, Julian Trouillon, Mylène Robert-Genthon, Benoît Gallet, Sylvie Elsen, Christine Moriscot, Suzan H. M. Rooijakkers, François Cretin, Ina Attrée.

**Methodology:** Manon Janet-Maitre, Stéphane Pont, Mylène Robert-Genthon, Benoît Gallet, Chantal Dumestre-Perard, Sylvie Elsen, Christine Moriscot, Bart W. Bardoel, Suzan H. M. Rooijakkers, François Cretin, Ina Attrée.

**Project administration:** Suzan H. M. Rooijakkers, Ina Attrée.

**Resources:** Stéphane Pont, Frerich M. Masson, Serena Sleiman, Mylène Robert-Genthon, Chantal Dumestre-Perard, Sylvie Elsen, François Cretin, Ina Attrée.

**Supervision:** Manon Janet-Maitre, Chantal Dumestre-Perard, Sylvie Elsen, Bart W. Bardoel, Suzan H. M. Rooijakkers, François Cretin, Ina Attrée.

**Validation:** Manon Janet-Maitre, Stéphane Pont, Bart W. Bardoel, Suzan H. M. Rooijakkers, François Cretin, Ina Attrée.

**Visualization:** Ina Attrée.

**Writing – original draft:** Ina Attrée.

**Writing – review & editing:** Manon Janet-Maitre, Stéphane Pont, Frerich M. Masson, Sylvie Elsen, Bart W. Bardoel, Suzan H. M. Rooijakkers, François Cretin, Ina Attrée.

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
