## [Decision Letter · Decision Letter 0]

18 Jul 2022

Dear Dr ATTREE,

Thank you very much for submitting your manuscript "Molecular features underlying Pseudomonas aeruginosa persistence in human plasma" for consideration at PLOS Pathogens. As with all papers submitted to the journal, yours was evaluated by a Section Editor in consultation with the Editorial Board. Your Article entitled " Molecular features underlying Pseudomonas aeruginosa persistence in human plasma” has also been evaluated by three independent reviewers, whose comments are attached. The reviewers appreciated the attention to an important problem, but raised some substantial concerns about the manuscript as it currently stands. Although one reviewer (referee #2) recommended to reject your manuscript, I share the point of view of the two others stating that your paper tries to address a crucial question demonstrating whether P. aeruginosa can persist in human plasma. While this paper represents a large body of work, it appears somehow build on new descriptions of three genes, which conferred to P. aeruginosa ability to persist in human plasma, but with no clear explanation whether they interplayed in the whole mechanism.

From the reports, you will see that while the referees found your work of some potential interest, they raise concerns about whether your findings are convincing. Please, note that I have decided to overcome the referee #2’s recommendation to reject the paper. Indeed, despite the present form need improvements and complementary data, I have the gut feeling that your study has potential to address the clinically relevant question about persistence of P. aeruginosa in human plasma.

In this paper, we understand that human plasma is a low ATP environment, which is a stress signal for P. aeruginosa that in turn forms polyP granules as phosphate storage to compensate low intracellular concentration of ATP. As a consequence, P. aeruginosa resists to MAC killing thanks to a membrane that is more resistant or that repairs faster.

As raised by the referees #1 and #3, the description of the mechanism deployed by P. aeruginosa to persist in human plasma lacks of a clear global picture allowing to connect all the pathways you have identified. Indeed, it is hard to eventually link the biotin biosynthesis pathway to the purin biosynthesis pathway to the srgABC operon. A final figure to sum up the mechanism of persistence of P. aeruginosa in human blood would have been very helpful and would strengthen your manuscript.

The major issues that I list below must be addressed before we would be willing to consider a revised version of your study. If you should take the referee’s comment into account and provide a point by point reply, I would appreciate if you can particularly pay attention to the minor revisions I have highlighted below. We cannot, of course, promise publication at that time. We therefore ask you to modify the manuscript according to the review recommendations before we can consider your manuscript for acceptance. Your revisions should address the specific points made by each reviewer.

The first major issue that I would like to reach to your attention is the mechanism that triggers phosphate storage granules in P. aeruginosa in response to ATP starvation. It is not very clear if the plasma is poor in ATP and therefore the bacterium is found to have low intracellular ATP or if the plasma is a stress signal that is detected by P. aeruginosa, which responds by modulating the intracellular synthesis of ATP. Indeed, the plasma supplementations with ATP restoring sensitivity seem in favor of the 1st hypothesis. However, on page 17 it is written “bacteria could use regulatory mechanisms to modulate purin and biotin synthesis…” which seems to explain that low intracellular level of ATP may result from a decrease biosynthesis in response to human plasma.

In other words, does low intracellular level of ATP result of uptake from poor ATP environment or of decreased synthesis in response to plasma?

The second major issue raised by reviewer #1 and #3 is the lack of clarity when linking membrane resistance to MAC with the low intracellular level of ATP to fully describe the mechanism of persistence of P. aeruginosa in plasma. How polyP granules and ATP starvation lead to MAC resistant membrane is very difficult to understand. Moreover the description of the Srg operon turns to be only described here without any role in the ATP starvation response described previously.

- How to connect ATP formation from polyP granules by Ppk and Srg in resistance to MAC killing?

- In BSI due to a WT bacterium, how the purin/biotin synthesis pathway as well as Srg are modulated since the evader phenotype is reversible and no genetic mutations occur?

Eventually, the Srg story seems very promising but very preliminary. Unless, you can connect the three pathways all in one mechanism explaining the plasma persistence deployed by P. aeruginosa in human blood, I guess your manuscript would benefit by focusing on one of the mutants. The ΔpurD is likely to be more straightforward in the attempt to describe the response upon ATP starvation (plasma stress signal, signal transduction and polyP granules, MAC killing membrane resistance).

I would like to raise your attention on some minor (because not contradicting the main message of your paper) issues.

1. Could you specify if your findings are also true for human blood and not only human plasma? Indeed, if it fully understandable that demonstration of persistence should have been made in plasma, it is important to link you results to that occurs in BSI.

2. Could you explain in the manuscript why you worked with ΔbioB and not mutants with clean deletions in other pur genes?

3. Could you explain in the manuscript why you worked with ΔpurD and not mutants with clean deletions in other bio genes?

4. Page 17 you stated that “although incapable of growth, ΔbioB and ΔpurD mutants survived and persisted in HIP”. Could you clarify in the manuscript if these mutants are incapable of growth in vitro or in HIP? In the first case, it implies that you should explain how you could grow such mutants before incubating them within HPI. In the second case, it implies that you should explain if there is killing or growth inhibition in HPI?

5. Page 19, second paragraph, could you be more specific when using “these bacteria”? For instance, it is difficult to understand if “these bacteria” wrote just after “In evaders” are the mutants mentioned at the beginning of the paragraph or P. aeruginosa that evaded to human plasma bactericidal activity in your experiments with WT or clinical strains.

We cannot make any decision about publication until we have seen the revised manuscript and your response to the reviewers' comments. Your revised manuscript is also likely to be sent to reviewers for further evaluation.

Sincerely,

Thomas Guillard, PharmD, PhD

Guest Editor

PLOS Pathogens

David Skurnik

Section Editor

PLOS Pathogens

Kasturi Haldar

Editor-in-Chief

PLOS Pathogens

orcid.org/0000-0001-5065-158X

Michael Malim

Editor-in-Chief

PLOS Pathogens

orcid.org/0000-0002-7699-2064

Reviewer's Responses to Questions

**Part I - Summary**

Reviewer #1: This study by Janet-Maitre et al. sheds light into potential mechanisms underlying P. aeruginosa persistence in human plasma. In an elaborate Tnseq screen, the authors identified several genes that play a role for establishment of a persistent subpopulation upon exposure of P. aeruginosa to plasma. Overall, the experiments were carefully executed and the manuscript is well written but would need some clarifications as pointed out below. Although no mechanisms was completely elucidated (which was stated as the goal in the first sentence in the discussion), the gene candidates are new and have not been reported before in relation to persistence. Potential weakness weaknesses include that the MS in its current form is more descriptive - it would benefit from more focus on one or two pathways.

Reviewer #2: In this paper, the authors successfully applied Tn-seq which is an excellent technique to study important genes of PA persistence in plasma and the authors successfully obtained a list of candidates from this screening. Unfortunately, the novelty is somewhat lacking because it is already common knowledge that PPK is relevant to PA virulence and survival in host from prior studies. It is thus not surprising to see in this paper that ∆PPK strains were not able to survive in plasma. Overall, I don't think there is strong ground to conclude that polyP is important to cope with plasma-related stress.

Here is another example of insufficient data for drawing their conclusions. Their study only confirmed that ATP levels could influence PA tolerance in one mutant (∆purD). While in the other mutant (∆bio), it seems that ATP levels are not important?

In "Analysis of isolated mutants" section they discuss but do not show their results? I just felt quite confused when I read this section. I am not sure if this should be in the result section if no data are provided.

Reviewer #3: Review "Molecular features underlying Pseudomonas aeruginosa persistence in human plasma" (PPATHOGENS-D-22-01021)

Janet-Maitre et al. described the potential and multiple mechanisms involved in persistence of P. aeruginosa in human plasma. The study is a nice approach of mechanistic explanations with novel results and other confirming previous publications. Authors raise the question of the mechanisms involved in persistence of P. aeruginosa in human plasma but without exposing a full explanation on discovered mechanisms. Some hypothesis would need more experimental approaches.

**Part II – Major Issues: Key Experiments Required for Acceptance**

Reviewer #1: While the reviewer sees this study as an important piece of work, it lacks somewhat some focus. The authors wrote "The aim of this study was to elucidate the mechanisms deployed by P. aeruginosa to persist in human plasma", however, they have not followed through with this goal. They have identified several pathways that appear to affect persistence in plasma, however, no direct mechanism is provided. In fact, the reviewer is wondering how a ∆purD strain, which is characterized by low ATP levels due to impaired purine biosynthesis, is able to produce such substantial amounts of polyP, which PPK can only generate from ATP.

Reviewer #2: See above

Reviewer #3: The emergence of so called “evaders” could result of mucoid overproduction, Psl production or type IV pili involvement. These data were already described in literature. In Janet-Maitre et al., described novel mechanisms: biotin and purine metabolisms and a new called serum resistance gene. The authors underlined the involvement of multiple and diverse mechanisms for human plasma persistence. However, even if the identification of these mechanisms is important, the link between the identified genes and persistence phenotype stays hypothetical. The investigation of serum factors triggering this reaction will be interested. Experiments on BSI isolates are needed to confirm the involvement of identified pathways. I am afraid that the mutants obtained in the “in vitro” conditions are not reflecting the in vivo conditions.

It is difficult to follow the different parts of the publication on purine, biotine and srgA pathways, as they have no link to each other. Authors should focus on one of this pathway to try to decipher the mechanisms involved with the use of double mutants for example. The focus could be on the formation of PolyP granules.

Concerning purine pathways, the finality will be the reservoir of energy under PolyP granules, and the decrease of ATP. However, the mechanisms involved are not clear.

About EDX experiments, the variations on osmium peak are puzzling between granules and cytoplasm but also between WT and mutant strain. It is definitely higher in the mutant and many others peaks that have appeared in granule area were not labelled (fig 5B). Is this attributable to higher lipid environment (e.g. for osmium) or increased count number on the area (e.g. as seen for copper signal that is intended to came from the grid). How many total counts were acquired for each condition? Even if one can agree with the authors that there is a presence of phosphorus, it is not reasonable to conclude on the quantity without better explanations on the standardization proces for EDX data acquisition.

Authors wrote:

1. “In the context of higher tolerance/persistence to antibiotics, it has been proposed that the ATP-depleted conditions could decrease overall protein synthesis, and thus availability of antibiotic targets (Shan et al., 2017). Similarly, slow growth per se, whatever the restrictions causing it, can trigger a persistent state (Pontes and Groisman, 2019). Although incapable of growth, ΔbioB and ΔpurD mutants survived and persisted in HIP for more than 24 hours.“

 But other mutants with slow growth or low ATP were not identified? It seems only purine and biotine dependant?

2. “Admittedly, mutations in the purine and biotin pathways are unlikely to occur in vivo, but bacteria could use regulatory mechanisms to modulate purine and biotin synthesis, to enter into an antibiotic- or complement system-tolerant state”

The other point is the link with what really happens in reality, in vivo. Why these mutants could not be isolated in vivo? Was the phenotype of real plasma evaders already studied? Could it possible to create mutant with induction/control of purine or biotine expression to confirm this hypothesis? Could it possible to evaluate these genes expression in evaders isolated from BSI?

3. “The number of plasma evaders present following incubation in plasma varies between experiments, strains and the plasma used ((Pont et al., 2020) and this work). Therefore, it is possible that the initial incubation which is considered poorly nutritive and stressful for bacteria may stochastically trigger the emergence of evaders.”

 Could subcultures in plasma be possible to avoid the impact of initial incubation?

4. “In evaders, in contrast, the outer membrane seems to have been effectively damaged by the MAC, suggesting that these bacteria have a more resistant envelope or can repair envelope damage more rapidly.”

 What are the mechanisms involved in MAC (only membrane impact?)?

 Comparison of evaders and mutants have to be done. Maybe by controlling /testing the fluidity of the membrane after MAC experiment.

**Part III – Minor Issues: Editorial and Data Presentation Modifications**

Reviewer #1: 1) The manuscript is written for an audience with direct expertise in the field. It would likely attract more readers if more background information was provided, e.g. it would benefit from a more thorough introduction into the complement system.

2) Page 7: "However, the screen did confirm the importance of LPS and more specifically OSA....". This was hard to find for the reviewer, the authors may want to help the reader which transposon mutants they are referring to.

3)Page 8: "This effect is of interest as Psl improved bacterial survival.....". It is also unclear hear, how retS/ladS and Psl relate to each other.

4) Page 11: survival increased ~20-fold according to Figure 2C. It would also be less confusing if the authors only use srgABC throughout the text as they do in the figures. They introduce the new names only in the 4th sub-panel.

5) Page 11: The authors claim that all thee srg genes are highly conserved in P. aeruginosa, which is not too surprising. Are there homologs in opther bacterial species and, if so, what are those?

6) Fig 2 E and F are not described in the text.

7) The authors decided to construct a ∆purD strain, why not a purM or purF mutants, as those genes had much higher Tn insertion frequencies?

8) Fig 4C: Any explanation why ATP and/or biotin addition doesn't affect the WT? Does the cell only take these compounds up when in desperate need? The reviewer also wonders why the replicate numbers differ throughout the experiments.

Reviewer #2: Abstract:

human plasma and blood – plasma is a component of blood

Author Summary:

biotin to significantly influence bacterial capacity to deal with - biotin significantly influence bacterial capacity to deal with

Page 5:

between 0.01 and 0.1% of evaders - between 0.01% and 0.1% of evaders

Page 7:

Random colony-picking failed to identify any LPS-related mutants - How many/percentage of colonies were picked up

Two isolated displayed hyper-mucoid phenotypes – Two isolated mutants?

Page 8:

Interestingly, bioinformatics analysis of the sequencing data revealed significant enrichment of transposon insertions in retS, which codes for a LadS antagonist (Table 1). This effect is of interest as Psl improved bacterial survival of a mucoid strain in serum (Jones and Wozniak, 2017). - First mention of Psl, need to elaborate. What’s the relationship between retS and Psl?

Our data thus suggest that either type IVb pili or CupE fimbriae could be also involved in P. aeruginosa plasma resilience, but the mechanisms involved need to be explored further. - which data, which figure?

Page 10:

Together, those results confirm the role of LPS, alginates, and Psl in resistance to the CS,

validating the Tn-seq approach - What results? How do they confirm that LPS, alginates and Psl are resistance to CS? and how to validate Tn-seq?

By the way, they might have forgotten to cite references in some sentences which is confusing since it is hard to know where the information is coming from.

Reviewer #3: 1. In result part:

“The size and number of the electron dense granules increased upon incubation of ΔpurD in human plasma (Fig. 5A, red arrows)”

The size depends on the site of the cut. It is therefore necessary to moderate this observation.

2. Screen or screening? Not sure of the term used.

3. I will propose a scheme for conclusion to clarify some points.

PLOS authors have the option to publish the peer review history of their article (what does this mean?). If published, this will include your full peer review and any attached files.

Reviewer #1: No

Reviewer #2: No

Reviewer #3: **Yes: **Fany REFFUVEILLE
---

## [Decision Letter · Decision Letter 1]

23 Nov 2022

Dear Dr ATTREE,

We are pleased to inform you that your manuscript 'Genome-wide screen in human plasma identifies multifaceted complement evasion of Pseudomonas aeruginosa' has been provisionally accepted for publication in PLOS Pathogens.

Best regards,

Thomas Guillard, PharmD, PhD

Guest Editor

PLOS Pathogens

David Skurnik

Section Editor

PLOS Pathogens

Kasturi Haldar

Editor-in-Chief

PLOS Pathogens

orcid.org/0000-0001-5065-158X

Michael Malim

Editor-in-Chief

PLOS Pathogens

orcid.org/0000-0002-7699-2064

Reviewer Comments (if any, and for reference):

Reviewer's Responses to Questions

**Part I - Summary**

Reviewer #1: My previous comments have been sufficiently addressed.

Reviewer #3: Janet-Maitre et al. have answered all raised questions. The modification of text, the addition of experiments and of a general scheme clarified all this work.

**Part II – Major Issues: Key Experiments Required for Acceptance**

Reviewer #1: My previous comments have been sufficiently addressed.

Reviewer #3: (No Response)

**Part III – Minor Issues: Editorial and Data Presentation Modifications**

Reviewer #1: My previous comments have been sufficiently addressed.

Reviewer #3: (No Response)

PLOS authors have the option to publish the peer review history of their article (what does this mean?). If published, this will include your full peer review and any attached files.

Reviewer #1: **Yes: **Jan-Ulrik Dahl

Reviewer #3: No

---

## [Editor Report · Acceptance letter]

20 Jan 2023

Dear Dr Attrée,

We are delighted to inform you that your manuscript, "Genome-wide screen in human plasma identifies multifaceted complement evasion of Pseudomonas aeruginosa," has been formally accepted for publication in PLOS Pathogens.

Best regards,

Kasturi Haldar

Editor-in-Chief

PLOS Pathogens

orcid.org/0000-0001-5065-158X

Michael Malim

Editor-in-Chief

PLOS Pathogens

orcid.org/0000-0002-7699-2064